# Optical detection of three modes of endocytosis at hippocampal synapses

Natali L Chanaday, Ege T Kavalali*

Department of Neuroscience, University of Texas Southwestern Medical Center, Dallas, United States

**Abstract** Coupling of synaptic vesicle fusion and retrieval constitutes a core mechanism ensuring maintenance of presynaptic function. Recent studies using fast-freeze electron microscopy and capacitance measurements reported an ultrafast mode of endocytosis operating at physiological temperatures. Here, using rat hippocampal neurons, we optically monitored single synaptic vesicle endocytosis with high time resolution using the vesicular glutamate transporter, synaptophysin and the V0a1 subunit of the vacuolar ATPase as probes. In this setting, we could distinguish three components of retrieval operating at ultrafast (~150–250 ms, ~20% of events), fast (~5–12 s, ~40% of events) and ultraslow speeds (>20 s, ~40% of events). While increasing $Ca^{2+}$ slowed the fast events, increasing temperature accelerated their time course. In contrast, the kinetics of ultrafast events were only mildly affected by these manipulations. These results suggest that synaptic vesicle proteins can be retrieved with ultrafast kinetics, although a majority of evoked fusion events are coupled to slower retrieval mechanisms.
DOI: https://doi.org/10.7554/eLife.36097.001

*For correspondence:
ege.kavalali@utsouthwestern.edu

Competing interests: The authors declare that no competing interests exist.

## Introduction

Synapses form the basic computational units in the brain as they transfer and process information with exquisite temporal and spatial precision. In presynaptic terminals, this is achieved through fast neurotransmitter release via fusion of synaptic vesicles within hundreds of microseconds upon the arrival of a presynaptic action potential (*Südhof, 2013*). In order to maintain synaptic performance during activity, synaptic vesicle proteins and lipids need to be retrieved and recycled. In contrast to the rapidity of the synaptic vesicle fusion process, the time course of endocytosis and its temporal relationship to presynaptic activity are widely debated. Early studies using electron microscopy (EM) and fluorescence imaging with FM dyes postulated retrieval and recycling of synaptic vesicles to occur within 30—90 s (*Miller and Heuser, 1984*; *Ryan et al., 1996*). However, subsequent experiments proposed a faster time course — spanning only about 2 s — for synaptic vesicle endocytosis based on capacitance measurements (*von Gersdorff and Matthews, 1994*), a time course also corroborated by FM dye-based studies (*Klingauf et al., 1998*; *Kavalali et al., 1999*). In recent years, experiments using fast-freeze EM (*Watanabe et al., 2013*) or improved capacitance measurements (*Delvendahl et al., 2016*) produced estimates within 100—500 ms for synaptic vesicle retrieval in central synapses. This so-called ultrafast synaptic endocytosis appeared to predominantly occur near physiological temperatures (~36°C) (*Watanabe and Boucrot, 2017*). In contrast, experiments using synaptic vesicle proteins labeled with pHluorin (a pH-sensitive derivative of GFP) to optically monitor single synaptic vesicle endocytosis gave rise to variable and often contradicting results (*Balaji and Ryan, 2007*; *Gandhi and Stevens, 2003*; *Granseth et al., 2006*; *Leitz and Kavalali, 2011*; *Zhu et al., 2009*) with the extra complication that these studies were typically conducted at room temperature.

In the present study, we aimed to monitor retrieval of single synaptic vesicles with improved time resolution at near physiological temperatures to resolve the discrepancies between fast EM and

**eLife digest** Nerve cells or neurons exchange information at junctions called synapses. To send a message to its neighbor, a neuron must release molecules called neurotransmitters into the synapse. These then bind to receptor proteins on the neighboring cell. But neurons do not release neurotransmitter molecules one at a time. Instead they release them in packages called vesicles. Each vesicle contains about 1,000 molecules, which it releases by fusing with the cell membrane. The entire process takes less than one thousandth of a second.

Synaptic vesicles are complex structures made up of many different proteins and lipids. To help ensure that neurons do not run out of vesicles, cells retrieve and recycle these components via a process called endocytosis. A number of studies have attempted to measure how long this retrieval process takes. But the studies – which used a variety of different techniques – yielded results ranging from a few hundredths of a second to more than a minute.

Chanaday and Kavalali have now resolved this discrepancy by using fluorescence microscopy to study the retrieval process in rat brain cells. By attaching a fluorescent tag to specific molecules within the vesicle membrane, Chanaday and Kavalali were able to track individual vesicles. The results revealed that neurons retrieve vesicles from synapses via three different pathways. At temperatures like those in the rodent or human body, an 'ultraslow' pathway takes more than 20 seconds to retrieve vesicles. By contrast, a 'fast' pathway takes about 5 to 12 seconds. The quickest option, an 'ultrafast' pathway, retrieves vesicles in about 150 to 250 milliseconds. Increasing the temperature speeds up the fast pathway but has no effect on the ultrafast pathway.

Neurons can thus retrieve vesicles from synapses in about 200 milliseconds, or one fifth of a second. Nevertheless, they retrieve about 80% of their vesicles using the two slower pathways. Identifying the mechanisms responsible for vesicle retrieval will help reveal how synapses work, as well as what can go wrong. Changes in components of synaptic vesicles contribute to several neurological and psychiatric diseases. Developing drugs that target synaptic vesicle recycling could be a promising therapeutic avenue.

DOI: https://doi.org/10.7554/eLife.36097.002

capacitance measurements and optical approaches. Besides enabling direct visualization of single vesicle retrieval kinetics, visualizing pHluorin-tagged synaptic vesicle protein dynamics provides the unique advantage of examining synaptic vesicle retrieval in a molecularly specific manner (*Kavalali and Jorgensen, 2014*). Under these conditions, using fluorescently tagged vesicular glutamate transporter, synaptophysin and V0a1 subunit of the vacuolar ATPase as probes, we detected three components of synaptic vesicle endocytosis: ultrafast (~150–250 ms), fast (~5–12 s) and ultra-slow (>20 s). This indicates that multiple endocytic pathways with different kinetics are operating at presynaptic terminals. Interestingly, while $Ca^{2+}$ and temperature were robust regulators of fast events, the kinetics of ultrafast events were relatively impervious to these manipulations. Overall, these results provide a bridge between dynamic optical measurements of synaptic vesicle protein retrieval and EM or capacitance based methods that monitor membrane trafficking.

## Results

### Monitoring endocytosis with vGluT1-pHluorin

The vesicular glutamate transporter 1 (vGluT1) bound to the pH sensitive derivative of GFP, pHluorin has been widely used to track synaptic vesicle trafficking based on its low plasma membrane expression and high signal-to-noise ratio (*Voglmaier et al., 2006*; *Balaji and Ryan, 2007*; *Leitz and Kavalali, 2011*). The pHluorin fluorescence is quenched in the acidic environment of the synaptic vesicle lumen and peaks after fusion with the plasma membrane (*Figure 1A*). Under strong stimulation (e.g. 40 Hz 5 s in *Figure 1A–C*), recapture of synaptic proteins from the plasma membrane results in a decay in fluorescence, reflecting synaptic vesicle retrieval and subsequent vesicle re-acidification (*Figure 1A*). At room temperature (~24°C), increasing extracellular $Ca^{2+}$ concentration from 2 mM to 8 mM leads to an increase in the number of fused vesicles (seen as an increase in peak amplitude) and a slight increase in the decay time, suggesting saturation of the endocytic machinery under

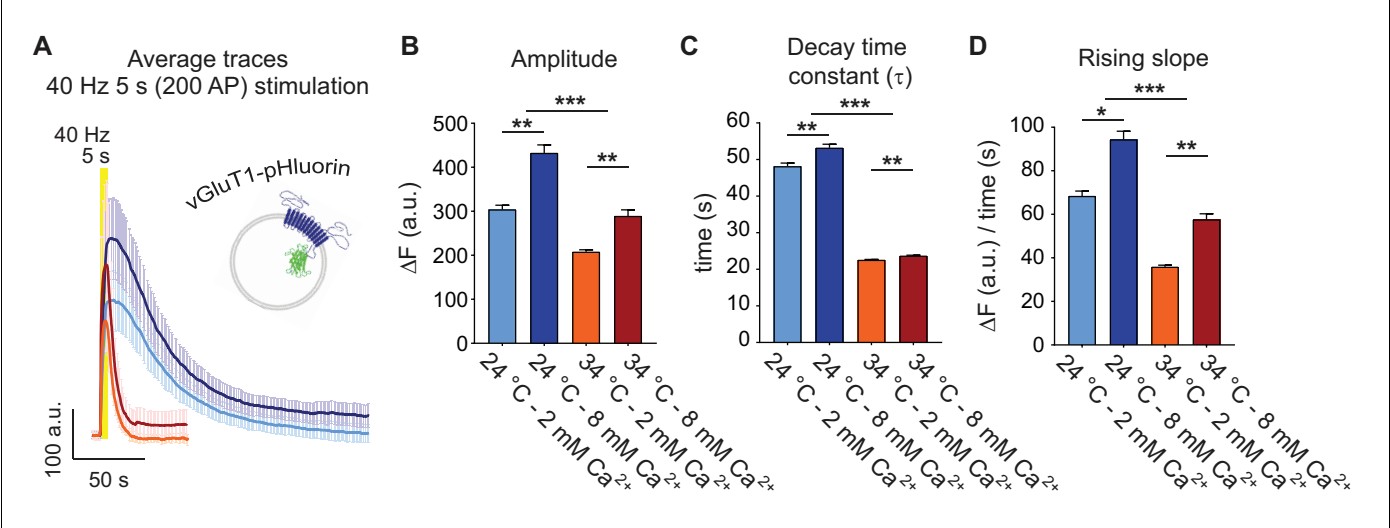

**Figure 1.** Physiological temperature synchronizes multivesicular release and accelerates bulk endocytosis, measured with vGluT1-pHluorin. (A) Average non-normalized 40 Hz (200 AP) traces of vGluT1-pHluorin at 24°C and 2 mM (light-blue) or 8 mM extracellular $Ca^{2+}$ (blue), or at 34°C and 2 mM (orange) or 8 mM $Ca^{2+}$ (red). (B) Amplitude of fluorescence responses after 40 Hz stimulation. (C) Average decay time constants ($\tau$) of the fluorescence return to baseline after 40 Hz stimulation, calculated with a single exponential decay fit. (D) Slope of the rise in fluorescence triggered by 40 Hz stimulation, expressed as change in fluorescence over time and calculated by linear regression. Statistical analysis was performed applying Kruskal-Wallis analysis (non-parametric ANOVA) with Dunn's multiple comparisons post-test. *p<0.05; **p<0.01; ***p<0.001; ***p<0.0001.

DOI: https://doi.org/10.7554/eLife.36097.003

strong, repetitive stimulation (*Figure 1A–C*). Earlier studies have demonstrated that at physiological temperatures (34–36°C) the time course of bulk synaptic vesicle endocytosis seen after strong stimulation shows accelerated kinetics (*Fernández-Alfonso and Ryan, 2004*; *Renden and von Gersdorff, 2007*; *Soykan et al., 2017*) possibly via activation of ultrafast endocytosis mechanisms (*Delvendahl et al., 2016*; *Watanabe et al., 2013*). In agreement with this premise, in our experiments, increasing the temperature from ~24 to 34°C led to a ~ 2.5 fold decrease in the fluorescence decay time constants after 40 Hz stimulation (*Figure 1C*). Moreover, at 34°C we observed faster fluorescence rise times at stimulation onset compared to room temperature (*Figure 1D*), consistent with an increase in the synchronicity of release seen in previous electrophysiological studies (*Pyott and Rosenmund, 2002*).

In the next set of experiments, we focused on single synaptic vesicle fusion and retrieval events induced during sparse low frequency stimulation (0.05 Hz). At the single vesicle level, fluorescence dwell times detected after fusion are representative of the time the pHluorin-tagged protein resides at the presynaptic plasma membrane before being retrieved. For action potential (AP) evoked release the kinetics of this process is negatively regulated by $Ca^{2+}$ concentration in a synaptotagmin-dependent manner (*Leitz and Kavalali, 2011*; *Li et al., 2017*). To visualize potential ultrafast retrieval of single synaptic vesicles, we monitored single fusion events at near physiological temperatures. Moreover, we increased the time resolution of our measurements, by a combination of faster image acquisition settings and time-domain de-noising of fluorescence signals based on methods originally developed to de-noise electrophysiological single channel recordings (*Chung and Kennedy, 1991*). This non-linear algorithm consists of a series of backward and forward predictors, similar to moving averages, used to calculate the probability of occurrence of fast changes in the signal, i.e. sharp changes in amplitude above the noise, in a Bayesian framework (*Figure 2A*). This type of noise reduction preserves the amplitude of the measured signal without major distortion of the edges compared to other classical methods based on the fast Fourier transform (see *Figure 2B* and *Figure 2—figure supplement 1*). As the filtering method we used assumes Gaussian distribution of the noise (*Chung and Kennedy, 1991*), we first validated that our experimental noise shows a Gaussian distribution (*Figure 2—figure supplement 1A–B*). These new settings enabled us to probe the fastest rate at which synaptic vesicle protein retrieval takes place at physiological temperatures. Noise reduction of the fluorescence traces from each region of interest (ROI) allowed us to detect

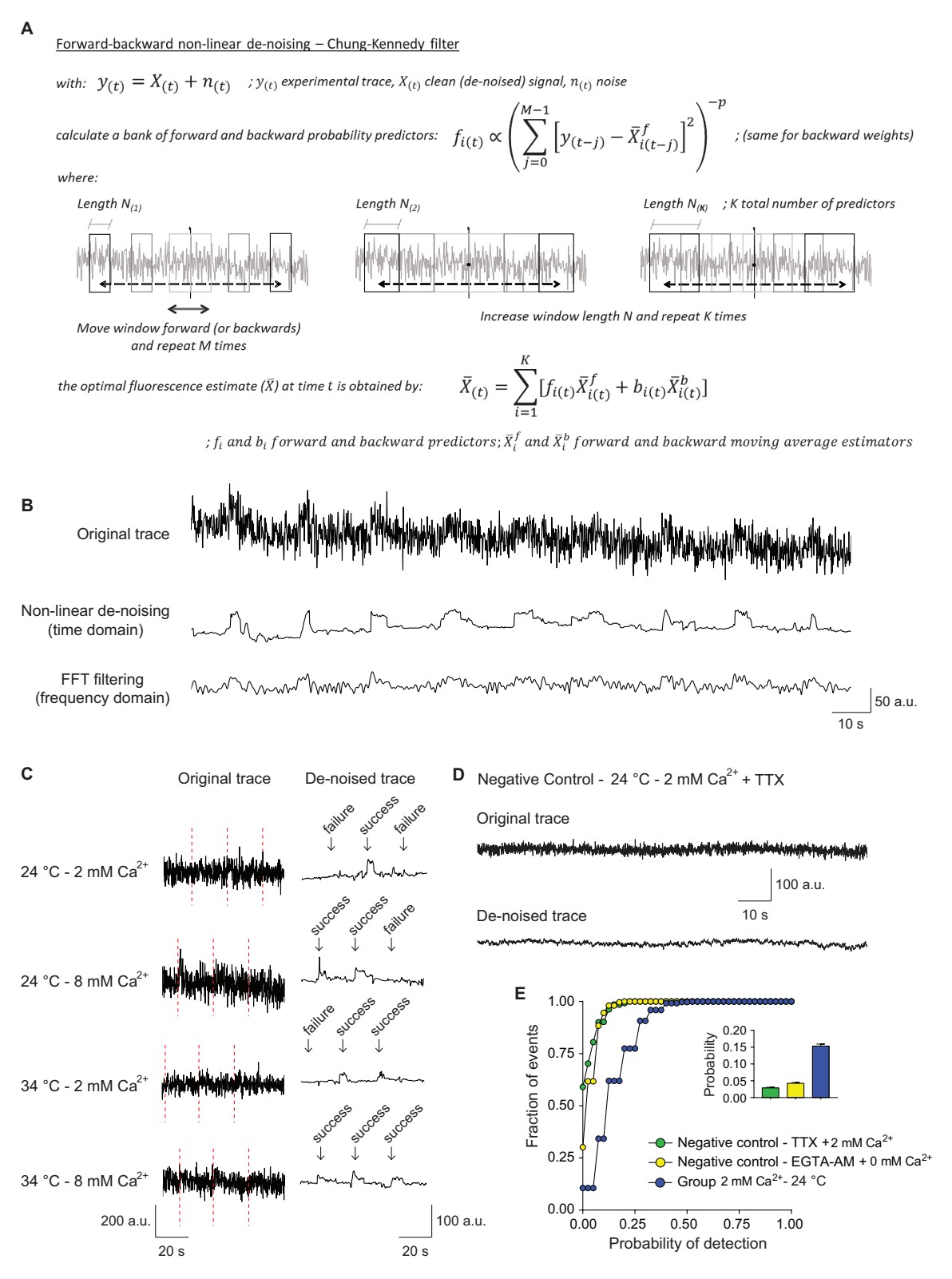

**Figure 2.** Signal de-noising increases the accuracy of detection and analysis of single synaptic vesicle fusion events. (**A**) Formulas and schematic representation of the Chung – Kennedy non-linear filter (for detailed explanation see *Chung and Kennedy, 1991*). (**B**) Example fluorescence trace before (top) and after processing using the Chung – Kennedy de-noising (middle) or the fast Fourier transform (FFT) filtering at 2 Hz (bottom). Appreciate the reduction in peak amplitude caused by FFT. (**C**) Comparison of original not processed vGluT1-pHluorin traces with the result signal after
*Figure 2 continued on next page*

*Figure 2 continued*

bleaching, background and noise corrections, at different extracellular $Ca^{2+}$ concentrations and temperatures. The failure analysis used to quantify probability of release is exemplified. (D) Example fluorescence trace over time, before and after de-noising, of a negative control bouton imaged at room temperature in the presence of tetrodotoxin (TTX). (E) Cumulative histogram of probability of release measured at room temperature, for positive control boutons (2 mM extracellular $Ca^{2+}$) and two negative control groups: TTX group corresponds to boutons imaged in the presence of TTX, 2 mM $Ca^{2+}$ and absence of electrical stimulation; EGTA-AM group corresponds to cultures pretreated with EGTA-AM 100 μM for 15 min in a 0 mM $Ca^{2+}$ medium, and posteriorly imaged at room temperature with 0 mM extracellular $Ca^{2+}$ and similar stimulation paradigm applied to the positive control group. The release probability for negative controls is <0.03, which corresponds to <20% of the mean release probability in the positive control group. Positive control: 368 boutons; Negative control – TTX: 401 boutons; Negative control – EGTA-AM: 260 boutons. All data are from 2 to 3 independent experiments. Also see *Figure 2—figure supplement 1* for controls regarding recovery and analysis of simulated traces.

DOI: https://doi.org/10.7554/eLife.36097.004

The following figure supplement is available for figure 2:

**Figure supplement 1.** Analysis of simulated fluorescence traces.

DOI: https://doi.org/10.7554/eLife.36097.005

small amplitude fusion events and reliably estimate release probability using failure analysis (*Figure 2C*). Moreover, dwell times of the fluorescent probe at the plasma membrane prior to retrieval could be determined with higher precision, enabling a clear discrimination between dwell times and the subsequent decay in fluorescence due to reacidification. Dwell time was defined as the total time after fusion that maintains maximum (peak) amplitude until the first derivative starts to be negative (due to re-acidification and the consequent decay of the signal). To estimate the accuracy of our event detection algorithm, we generated negative imaging controls with no stimulation in the presence of TTX or in 0 mM extracellular $Ca^{2+}$ following EGTA-AM treatment to buffer intracellular $Ca^{2+}$. In these control experiments, we detected events with a probability of ~0.03% or less (likely reflecting false positives along with spontaneous fusion events), well within acceptable boundaries (~5 fold less than typical event detection rate under normal conditions) (*Figure 2D–E*). To assess the precision of the dwell time calculation, artificial traces with different release probabilities and dwell time distributions were simulated and similarly de-noised and analyzed. This analysis corroborated the limited contribution of false positives to the dwell time durations obtained (*Figure 2—figure supplement 1C–E*). In conclusion, all these experimental and simulated controls enabled us to tune the de-noising parameters to analyze the duration of dwell times in the experimental traces (*Chung and Kennedy, 1991*).

Example vGluT1-pHluorin fluorescence traces of single synaptic vesicle fusion events after de-noising are depicted in *Figure 3A*. As shown in *Figure 3B and C*, mean release probability at 2 mM extracellular $Ca^{2+}$ is ~0.15, in agreement with previous reports (*Leitz and Kavalali, 2011*; *Murthy et al., 1997*). Increasing $Ca^{2+}$ concentration to 8 mM leads to a marked increase in release probability (*Figure 3B–C*), although switching from room temperature (~24°C) to 34°C, did not result in a major change in release probability, consistent with previous electrophysiological measurements (*Kushmerick et al., 2006*; *Pyott and Rosenmund, 2002*). Initially, we detected that 40 ± 2% of all fusion events did not return to baseline during the course of imaging (>20 s, and see *Figure 3H*) suggesting an ultraslow pathway for endocytosis (*Gandhi and Stevens, 2003*; *Zhu et al., 2009*). The remaining events (~60%) showed dwell times of variable lengths which displayed a non-normal distribution (*Figure 3D–G*) with a peak at ~250 ms. Further analysis of the histograms revealed that they are best represented by a double exponential decay function, pointing to the co-existence of two endocytic mechanisms with different mean rates. The fitting of dwell time distributions for the experimental groups revealed an ultrafast endocytic component with an average time course of about 150–250 ms, and a fast process that proceeds with an order of magnitude slower time course of around 5–12 s (arrows in *Figure 3D–G*). Increasing extracellular $Ca^{2+}$ concentration from 2 to 8 mM led to only a 10–15% slowdown of ultrafast endocytosis at both temperatures (*Figure 3D–G*). The fast component of endocytosis, on the other hand, showed a steeper temperature dependence. Increasing extracellular calcium concentration from 2 to 8 mM led to a ~ 60% slowdown of fast endocytosis at 24°C, while increasing calcium at 34°C led to only a ~ 10% slowdown (*Figure 3D–G*). This result indicates that increasing temperature triggers an overall acceleration of vGluT1-pHluorin retrieval mainly through regulation of the fast component of endocytosis, and this form of regulation is more evident at higher $Ca^{2+}$ concentrations (compare insets in *Figure 3E and G*). Kolmogorov-

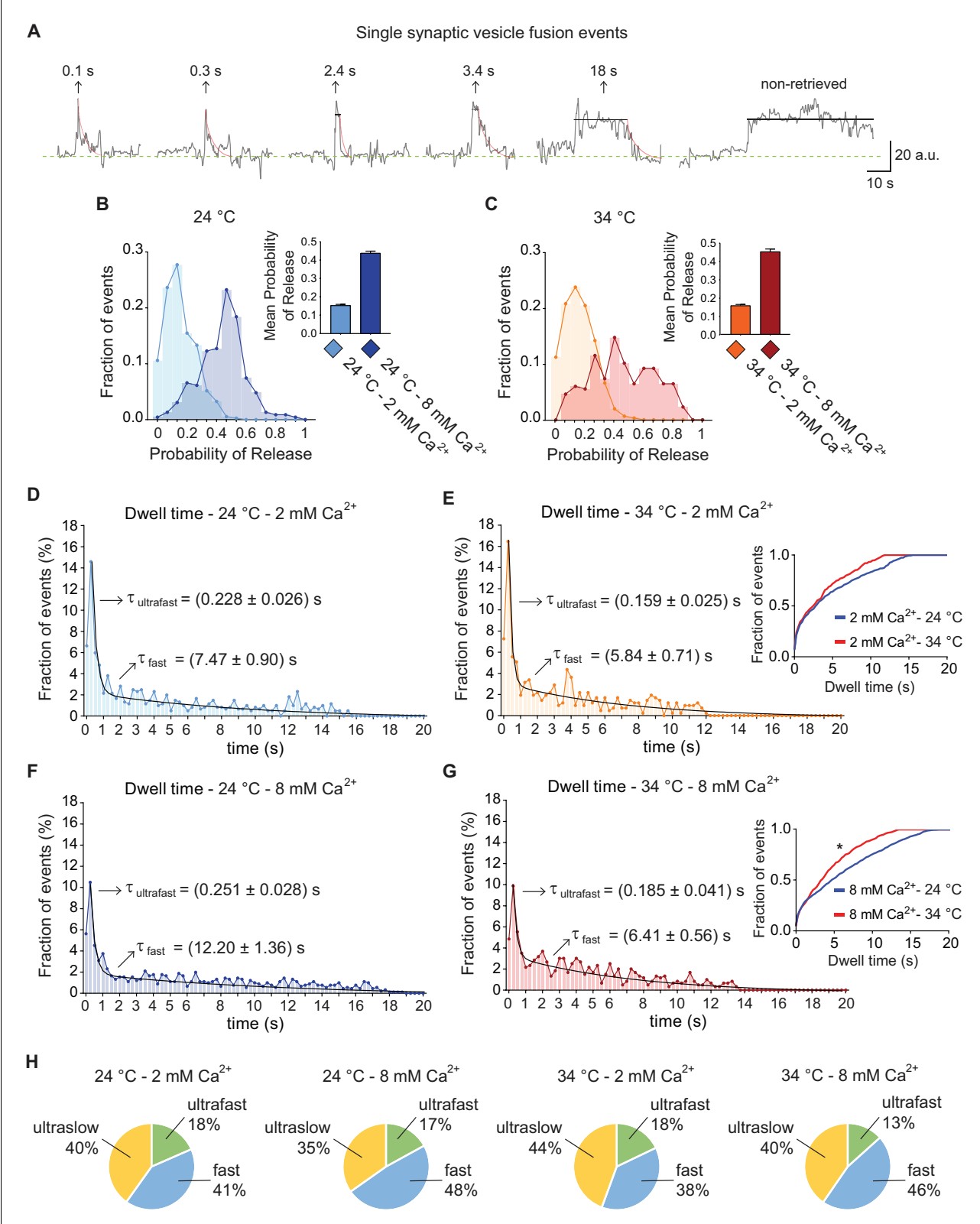

**Figure 3.** vGluT1-pHluorin reveals that endocytosis after single AP stimulation can occur through three kinetically distinct pathways. (**A**) Example traces of de-noised single synaptic vesicle fusion events measured with vGluT1-pHluorin, showing different dwell time lengths. (**B**) Distribution of probabilities of release at 24°C with 2 mM (light-blue) or 8 mM (blue) extracellular $Ca^{2+}$. Inset: mean release probability for each group. 2 mM $Ca^{2+}$ 24°C: 0.15 ± 0.01. 8 mM $Ca^{2+}$ 24°C: 0.44 ± 0.01. (**C**) Distribution of probabilities of release at 34°C with 2 mM (orange) or 8 mM (red) $Ca^{2+}$. Inset: mean release probability

*Figure 3 continued on next page*

*Figure 3 continued*

for each group. 2 mM $Ca^{2+}$ 34°C: 0.16 ± 0.01. 8 mM $Ca^{2+}$ 34°C: 0.45 ± 0.02. Note that temperature does not affect release probability in the range tested. (D) Distribution of dwell time durations at 24°C and 2 mM extracellular $Ca^{2+}$. Black line: double exponential decay fit (R-square = 0.921; RRS = 0.001741). Arrows: decay constants for the ultrafast and fast components of the exponential. (E) Distribution of dwell time durations at 34°C and 2 mM extracellular $Ca^{2+}$. Black line: double exponential decay fit (R-square = 0.912; RRS = 0.002393). Arrows: decay constants for the ultrafast and fast component of the exponential. Inset: cumulative histogram comparing the effect of temperature on dwell times at 2 mM $Ca^{2+}$, there are not significant differences. (F) Distribution of dwell time durations at 24°C and 8 mM extracellular $Ca^{2+}$. Black line: double exponential decay fit (R-square = 0.921; RRS = 0.001059). Arrows: decay constants for the ultrafast and fast component of the exponential. (G) Distribution of dwell time durations at 34°C and 8 mM extracellular $Ca^{2+}$. Black line: double exponential decay fit (R-square = 0.899; RRS = 0.001407). Arrows: decay constants for the ultrafast and fast component of the exponential. Inset: cumulative histogram comparing the effect of temperature on dwell times at 8 mM $Ca^{2+}$, *=p < 0.05. (H) Pie charts depicting the relative contribution of the three modes of retrieval to total endocytosis for all experimental groups. Green: ultrafast retrieval (dwell time duration between 0 and 1 s). Blue: fast endocytosis (dwell time of 1 to 20 s). Yellow: ultra-slow retrieval (>20 s). Note that the percentage of each type of endocytosis is not greatly affected by changes in $Ca^{2+}$ or temperature. For D to G. Kolmogorov-Smirnov test: 24°C 2 mM $Ca^{2+}$ vs. 24°C 8 mM $Ca^{2+}$: p=0.0026; 34°C 2 mM $Ca^{2+}$ vs. 34°C 8 mM $Ca^{2+}$: p=0.0043; 24°C 2 mM $Ca^{2+}$ vs 34°C 2 mM $Ca^{2+}$: p=0.0265; 24°C 8 mM $Ca^{2+}$ vs 34°C 8 mM $Ca^{2+}$: p=0.0015. Kruskal-Wallis test: p<0.0001; Dunn's post-test: 24°C 2 mM $Ca^{2+}$ vs. 24°C 8 mM $Ca^{2+}$: p<0.0001; 34°C 2 mM $Ca^{2+}$ vs. 34°C 8 mM $Ca^{2+}$: p<0.05; 24°C 2 mM $Ca^{2+}$ vs 34°C 2 mM $Ca^{2+}$: not-significant; 24°C 8 mM $Ca^{2+}$ vs 34°C 8 mM $Ca^{2+}$: p<0.0001. For all the data presented in this figure: 24°C – 2 mM $Ca^{2+}$: 576 boutons from eight coverslips; 24°C – 8 mM $Ca^{2+}$: 496 boutons from seven coverslips; 34°C – 2 mM $Ca^{2+}$: 442 boutons from 10 coverslips; 34°C – 8 mM $Ca^{2+}$: 370 boutons from nine coverslips. At least three independent experiments (cultures).

DOI: https://doi.org/10.7554/eLife.36097.006

Smirnov pair-wise comparison of cumulative distributions, as well as Kruskal-Wallis non-parametric analysis of variances, further corroborated that $Ca^{2+}$ concentration and temperature had a significant effect on the shift of the distributions (*Figure 3D–G*, legend). While the mean speed of endocytosis for both fast and ultrafast pathways was regulated by temperature and $Ca^{2+}$, the relative contribution of each mode to total protein retrieval did not change significantly (*Figure 3H*). Overall under all conditions around 40% of fusion events were not followed by endocytosis in the imaging period (>20 s), 13–18% undergo ultrafast endocytosis and 38–48% of vGluT1-pHluorins were retrieved through a slow pathway (*Figure 3H*). The mean amplitude of fusion events was similar for all types of retrieval (data not shown). In addition, to test whether the two components of retrieval we identified were a consequence of denoising, we averaged non-denoised traces in an unbiased manner and found two phases of fluorescence decay with ultrafast (~250 ms) and fast kinetics (~3–7 s) (*Figure 4A*). Although, this analysis does not distinguish dwell time and re-acidification kinetics, it supports our conclusion that two kinetically distinct processes retrieve vesicle proteins with ultrafast and fast speeds. Interestingly, in neurons where the $Ca^{2+}$ sensor synaptotagmin-1 (syt1) was knocked-down (KD) the fast component is greatly reduced and vGluT1-pHluorin is retrieved almost exclusively by ultrafast endocytosis (70–80% of all fusion events, *Figure 4B–C*), suggesting that ultrafast endocytosis is syt1-independent (see *Li et al., 2017*). Moreover, ultraslow retrieval (>20 s) is also abolished in syt1 KD neurons (*Figure 4C*). This was not the case for synaptotagmin-7 (syt7) KD neurons, where averaged single vesicle traces showed a double exponential decay ($\tau_1 = 0.31 \pm 0.05$ s, $\tau_2 = 8.5 \pm 4.8$ s) similarly to control synapses. Taken together, these results indicate that the speed of fast single vesicle retrieval is markedly accelerated by temperature and slowed down by extracellular $Ca^{2+}$ concentration, possibly through syt1, whereas the kinetics of ultrafast events are less sensitive to these factors.

Approximately 60% of total fusion events displayed dwell times and subsequent decay. Among these, only in 50% the fluorescence returned back to baseline completely, consistent with full quantal retrieval of vGluT1-pHluorins (fractional retrieval of 1.0 ± 0.2, see *Figure 4D–E*). Approximately, 35% of the fusion events that were followed by a measurable dwell time, showed partial retrieval, indicating that some vGluT1-pHluorin molecules might diffuse away from the site of fusion and are temporarily unavailable for endocytosis. The remaining ~15% of events showed excess retrieval, suggesting that more vGluT1-pHluorin molecules were retrieved compared to the ones that fused (*Figure 4D–E*), similar to what was described in other systems (e.g. *Chung and Kavalali, 2009*; *Van Hook and Thoreson, 2012*; *Zhu et al., 2009*). Plotting the fraction of retrieval for each fusion event as a function of the duration of its dwell time revealed that the dispersion of retrieval values increased with longer dwell times (*Figure 4F*). While the dispersion was ±20% for ultrafast events (dwell times of 0.0 to 0.5 s), for fast events dispersion increased from approximately ±40% to around ±80–90% as the values of dwell times increased (see legend of *Figure 4F*). This increasing variability in the

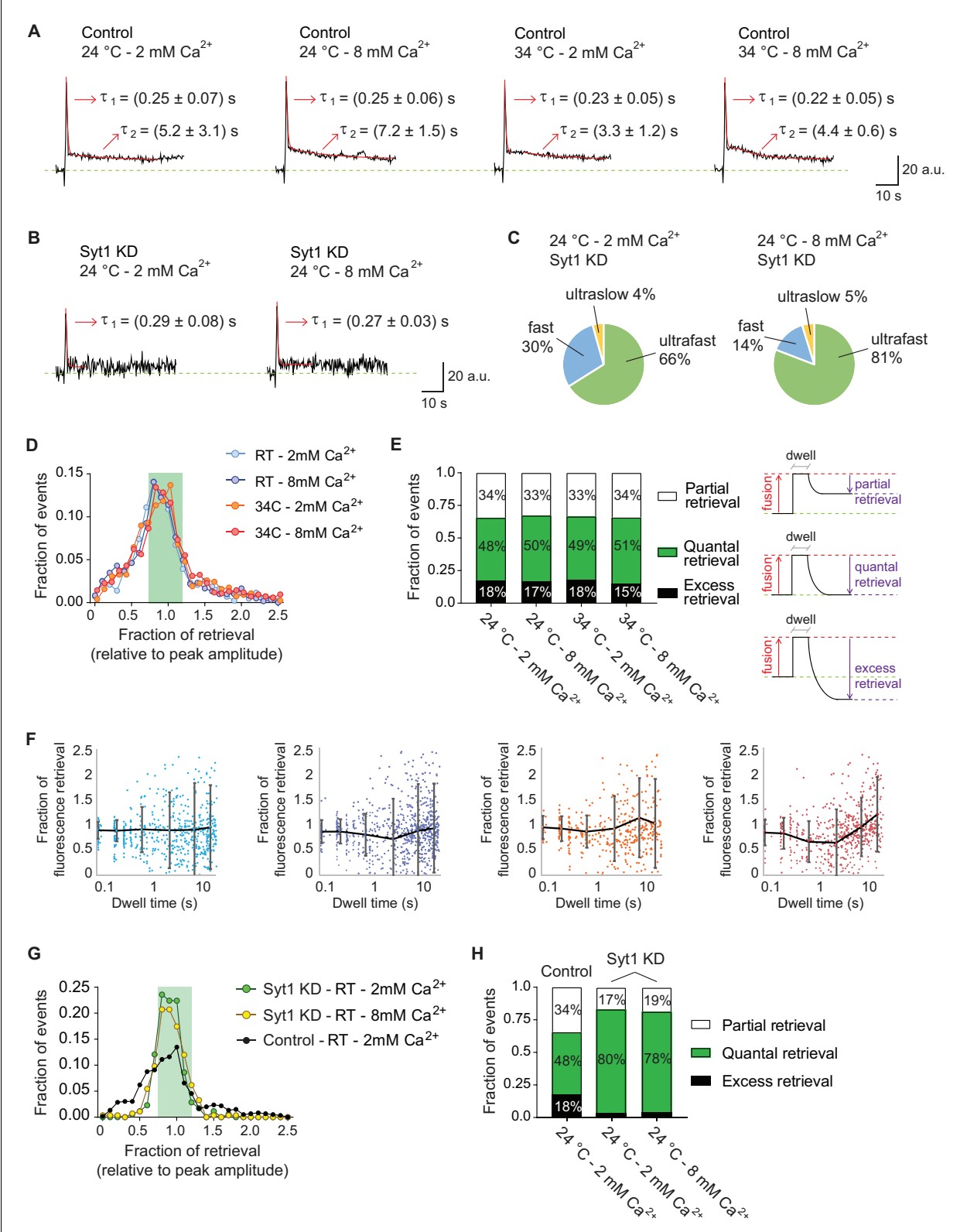

**Figure 4.** Ultrafast endocytosis is associated with quantal release, while fast endocytosis shows a wide variability in the fraction of retrieved vGluT1-pHluorin. (**A**) Average of all fusion events measured with vGluT1-pHluorin at 24 and 34°C, and at 2 or 8 mM extracellular $Ca^{2+}$ (N of events, boutons and experiments correspond to the ones presented in *Figure 3*). Red lines show the exponential fit, with two phases of decay: one ultrafast ($\tau_1$) and one fast ($\tau_2$). Arrows: values of the decay constants for the ultrafast and fast components. Note that values agree with the timescales obtained by fitting the

*Figure 4 continued on next page*

*Figure 4 continued*

distributions of individual dwell time durations (non-averaged). (B) Averaged traces of fusion events measured in syt1 KD hippocampal neurons, in 2 or 8 mM extracellular $Ca^{2+}$. Red lines show the exponential fit, with only the ultrafast component present ($\tau_1$). Arrows: values of the decay constants. (C) Pie charts depicting the relative contribution of the three modes of retrieval to total endocytosis in syt1 KD neurons, measured using vGluT1-pHluorin. Green: ultrafast retrieval (dwell time duration between 0 and 1 s). Blue: fast endocytosis (dwell time of 1 to 20 s). Yellow: ultra-slow retrieval (>20 s). Note that 70–80% of endocytosis is mediated by an ultrafast mechanism in syt1 KD neurons. (D) Histogram showing the distribution of the level of retrieval (relative to the fluorescence amplitude of the fusion), showing similar distribution for all tested conditions in wild-type neurons. (E) Left: bar graph showing the percentage of each type of retrieval contributing to total retrieval. Note that regardless of $Ca^{2+}$ concentration or temperature around 50% of fusion events are followed by quantal retrieval (fraction of retrieval in the range of 1.0 ± 0.2). Right: schematic representation of how the fraction of retrieval was calculated and the three types of retrieval found: partial retrieval (<0.8), quantal retrieval (1.0 ± 0.2), excess retrieval (>1.2). (F) Plots of fraction of retrieval as a function of dwell time duration, for 24 and 34°C at 2 or 8 mM extracellular $Ca^{2+}$. Note the increase in the dispersion of fraction of retrieval as dwell time length increases. Dots: individual values from each fusion event. Black lines and error bars: average fraction of retrieval and standard deviation for the following groups. 2 mM $Ca^{2+}$ at 24°C: 0.0–0.1 s dwell time – 0.9 ± 0.2 fraction of retrieval; 0.1–0.5 s dwell time – 0.9 ± 0.2 fraction of retrieval; 0.5–1.0 s dwell time – 0.9 ± 0.4 fraction of retrieval; 1–5 s dwell time – 0.9 ± 0.7 fraction of retrieval; 5–10 s dwell time – 0.9 ± 0.9 fraction of retrieval; 10–20 s dwell time – 0.9 ± 0.8 fraction of retrieval. 8 mM $Ca^{2+}$ at 24°C: 0.0–0.1 s dwell time – 0.9 ± 0.2 fraction of retrieval; 0.1–0.5 s dwell time – 0.9 ± 0.2 fraction of retrieval; 0.5–1.0 s dwell time – 0.8 ± 0.4 fraction of retrieval; 1–5 s dwell time – 0.7 ± 0.8 fraction of retrieval; 5–10 s dwell time – 0.9 ± 0.9 fraction of retrieval; 10–20 s dwell time – 0.9 ± 0.8 fraction of retrieval. 2 mM $Ca^{2+}$ at 34°C: 0.0–0.1 s dwell time – 0.9 ± 0.2 fraction of retrieval; 0.1–0.5 s dwell time – 0.9 ± 0.2 fraction of retrieval; 0.5–1.0 s dwell time – 0.8 ± 0.3 fraction of retrieval; 1–5 s dwell time – 0.9 ± 0.6 fraction of retrieval; 5–10 s dwell time – 1.0 ± 0.8 fraction of retrieval; 10–20 s dwell time – 1.0 ± 0.9 fraction of retrieval. 8 mM $Ca^{2+}$ at 34°C: 0.0–0.1 s dwell time – 0.8 ± 0.3 fraction of retrieval; 0.1–0.5 s dwell time – 0.8 ± 0.3 fraction of retrieval; 0.5–1.0 s dwell time – 0.7 ± 0.4 fraction of retrieval; 1–5 s dwell time – 0.6 ± 0.7 fraction of retrieval; 5–10 s dwell time – 0.9 ± 0.6 fraction of retrieval; 10–20 s dwell time – 1.2 ± 0.7 fraction of retrieval. (G) Distribution of the level of retrieval (relative to the fluorescence amplitude of the fusion), showing a sharper distribution for syt1 KD neurons compared to wild type (control) neurons. (H) Bar graph showing the percentage of each type of retrieval contributing to total retrieval. Around 80% of fusion events are followed by quantal retrieval in syt1 KD, at either 2 or 8 mM extracellular $Ca^{2+}$, contrasting with only 50% in control neurons. For B, C, G and H. Syt1 KD at 24°C and 2 mM $Ca^{2+}$: 120 events from eight coverslips; Syt1 KD at 24°C and 8 mM $Ca^{2+}$: 124 events from 14 coverslips. 4–5 independent experiments (cultures).

DOI: https://doi.org/10.7554/eLife.36097.007

retrieved fraction occurred similarly for all the temperatures and extracellular $Ca^{2+}$ concentrations tested. This result implies that as endocytosis gets slower, the probability of having partial retrieval of synaptic vesicle proteins or of having retrieval of double amount of protein compared to the amount that fused increases. As mentioned before, for neurons lacking syt1 the majority of fusion events were followed by ultrafast endocytosis, for these events vGluT1-pHluorin is endocytosed very efficiently, with ~80% of vGluT1-pHluorin fusion events showing quantal retrieval (1.0 ± 0.2, *Figure 4G–H*). Our results suggest that ultrafast endocytosis is $Ca^{2+}$, syt1 and temperature independent, and it rapidly retrieves approximately the same amount of protein that fused.

## Ultrafast and fast retrieval of synaptophysin and V0a1 subunit of the v-ATPase

Distinct synaptic vesicle proteins are typically coupled to diverse endocytic mechanisms (*Chanaday and Kavalali, 2017*; *Li et al., 2017*; *Pan et al., 2015*; *Voglmaier and Edwards, 2007*) Therefore, to evaluate the general validity of our results, we tested the trafficking of synaptophysin-1 fused to the red pH sensitive protein pHTomato (Syp1-pHTomato, *Figure 5A*) (*Li and Tsien, 2012*) under the same conditions. At 40 Hz stimulation, the increase of temperature from 24 to 34°C caused a ~ 2.5 fold rise in the bulk endocytic rate of Syp1-pHTomato (*Figure 5B*) coupled with a faster signal rise time (*Figure 5C*), consistent with our observations using vGluT1-pHluorin. At the single vesicle level, as before, we did not detect a change in release probability in response to increase in temperature (~0.15 at both tested temperatures) (*Figure 5D*). Sample traces of single synaptic vesicle fusion events detected with Syp1-pHTomato are shown in *Figure 5E*. Histograms of detectable dwell times (50 ± 2% of events) were best fitted with a double exponential, revealing the presence of two parallel endocytic processes for synaptophysin (*Figure 5F–I*). The mean time courses of these two processes were about 175–300 ms and 5–11 s, comparable to our previous results with vGluT1-pHluorin (arrows in *Figure 5F–I*). While the increase in temperature did not produce a significant effect in the overall endocytic rate at 2 mM $Ca^{2+}$ (*Figure 5F–G*, and inset in 5G), it caused an almost two-fold acceleration of retrieval at 8 mM extracellular $Ca^{2+}$ (*Figure 5H–I*, and inset in 5I). As observed for vGluT1-pHluorin, even though both ultrafast and fast processes are

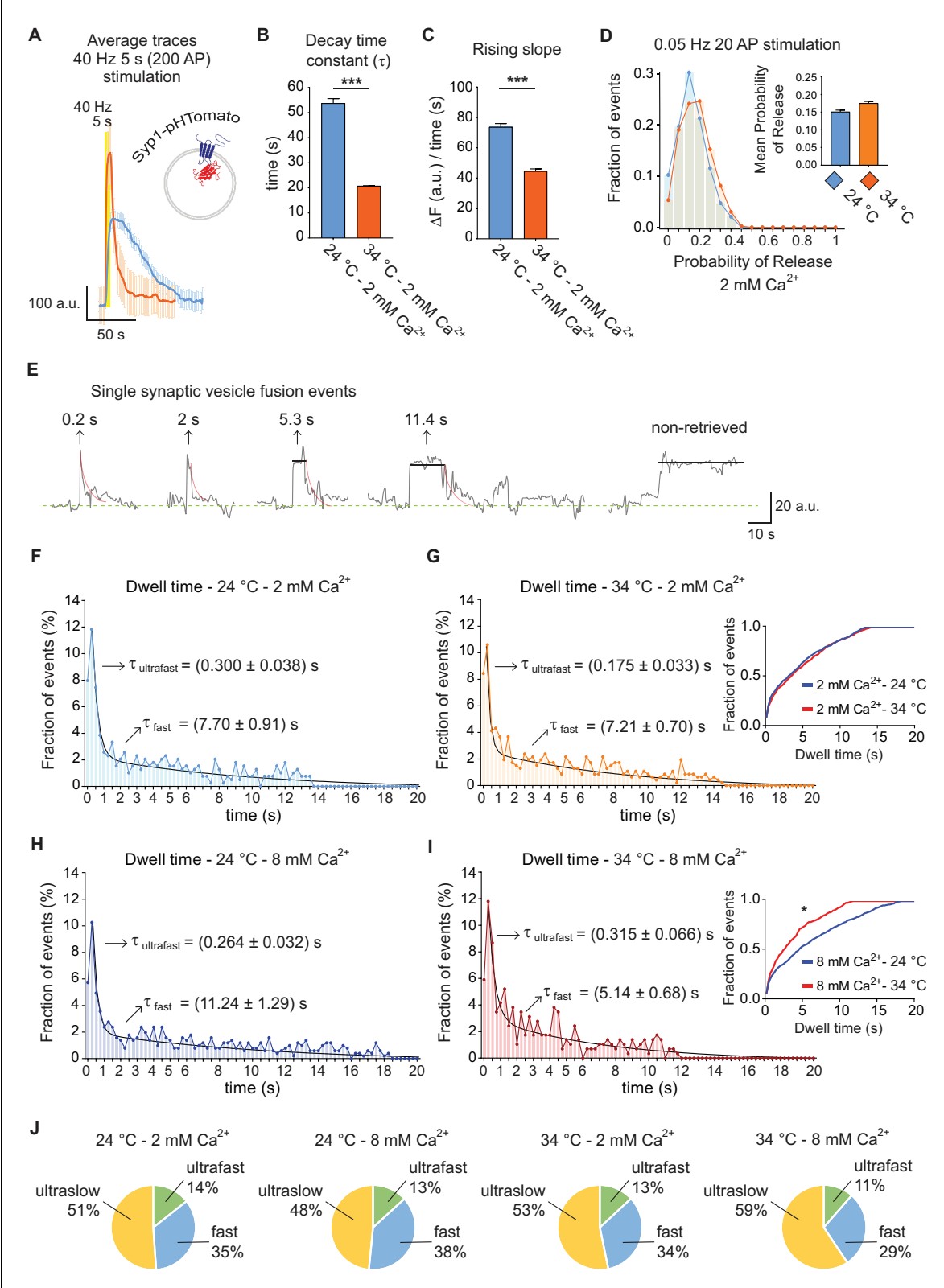

**Figure 5.** Trafficking of Syp1-pHTomato at different temperatures reproduces vGluT1-pHluorin behavior. (**A**) Average non-normalized 40 Hz (200 AP) traces of Syp1-pHTomato at 24°C and 2 mM extracellular $Ca^{2+}$ (blue), or at 34°C and 2 mM $Ca^{2+}$ (orange). (**B**) Average decay time constants ($\tau$) of the fluorescence return to baseline after 40 Hz stimulation, calculated with a single exponential decay fit. (**C**) Slope of the rise in fluorescence triggered by 40 Hz stimulation, expressed as change in fluorescence over time and calculated by linear regression. For B to D. Statistical analysis was performed

*Figure 5 continued on next page*

*Figure 5 continued*

applying Kruskal-Wallis analysis (non-parametric ANOVA) with Dunn's multiple comparisons post-test. *p<0.05; **p<0.01; ***p<0.001; ***p<0.0001. (**D**) Distribution of probabilities of release at 24°C (blue) or 34°C (orange) and 2 mM extracellular $Ca^{2+}$ concentration. Inset: mean release probability for each group. 2 mM $Ca^{2+}$ at 24°C: 0.15 ± 0.01. 2 mM $Ca^{2+}$ at 34°C: 0.17 ± 0.01. (**E**) Example traces of de-noised single synaptic vesicle fusion events measured with Syp1-pHTomato, showing different dwell time lengths. (**F**) Distribution of dwell time durations at 24°C and 2 mM extracellular $Ca^{2+}$. Black line: double exponential decay fit (R-square = 0.922; RRS = 0.001336). Arrows: decay constants for the ultrafast and fast component of the exponential. (**G**) Distribution of dwell time durations at 34°C and 2 mM extracellular $Ca^{2+}$. Black line: double exponential decay fit (R-square = 0.593; RRS = 0.001423). Arrows: decay constants for the ultrafast and fast component of the exponential. Inset: cumulative histogram comparing the effect of temperature on dwell times at 2 mM $Ca^{2+}$, there are no significant differences. (**H**) Distribution of dwell time durations at 24°C and 8 mM extracellular $Ca^{2+}$. Black line: double exponential decay fit (R-square = 0.911; RRS = 0.00122). Arrows: decay constants for the ultrafast and fast component of the exponential. (**I**) Distribution of dwell time durations at 34°C and 8 mM extracellular $Ca^{2+}$. Black line: double exponential decay fit (R-square = 0.865; RRS = 0.002644). Arrows: decay constants for the ultrafast and fast component of the exponential. Inset: cumulative histogram comparing the effect of temperature on dwell times at 8 mM $Ca^{2+}$, *=p < 0.05. (**J**) Pie charts showing the percentage of each mode of retrieval for all experimental groups. Green: ultrafast retrieval (dwell time duration between 0 and 1 s). Blue: fast endocytosis (dwell time of 1 s to 20 s). Yellow: ultra-slow retrieval (>20 s). Note that the ratio of each type of endocytosis does not change with $Ca^{2+}$ or temperature. For H to K. Kolmogorov-Smirnov test: 24°C 2 mM $Ca^{2+}$ vs. 24°C 8 mM $Ca^{2+}$: p=0.0011; 34°C 2 mM $Ca^{2+}$ vs. 34°C 8 mM $Ca^{2+}$: p=0.0015; 24°C 2 mM $Ca^{2+}$ vs 34°C 2 mM $Ca^{2+}$: non-significant; 24°C 8 mM $Ca^{2+}$ vs 34°C 8 mM $Ca^{2+}$: p<0.0001. Kruskal-Wallis test: p<0.0001; Dunn's post-test: 24°C 2 mM $Ca^{2+}$ vs. 24°C 8 mM $Ca^{2+}$: p<0.001; 34°C 2 mM $Ca^{2+}$ vs. 34°C 8 mM $Ca^{2+}$: non-significant; 24°C 2 mM $Ca^{2+}$ vs 34°C 2 mM $Ca^{2+}$: non-significant; 24°C 8 mM $Ca^{2+}$ vs 34°C 8 mM $Ca^{2+}$: p<0.0001. For all the data presented in this figure. 24°C – 2 mM $Ca^{2+}$: 423 boutons from seven coverslips; 24°C – 8 mM $Ca^{2+}$: 384 boutons from seven coverslips; 34°C – 2 mM $Ca^{2+}$: 393 boutons from six coverslips; 34°C – 8 mM $Ca^{2+}$: 361 boutons from seven coverslips. At least three independent experiments (cultures).

DOI: https://doi.org/10.7554/eLife.36097.008

regulated by $Ca^{2+}$ and temperature, the effect of these manipulations are more striking for the fast component and at 8 mM $Ca^{2+}$. Statistical analysis confirmed this finding (*Figure 5F–I*, legend). In contrast, the charts in *Figure 5J* show that there is only a modest effect of temperature and $Ca^{2+}$ in the proportion of the different modes of endocytosis after single AP stimulation. Our results so far revealed the co-existence of two rapid endocytic processes at hippocampal synapses (ultrafast and fast), whose speed is differentially regulated by $Ca^{2+}$ and temperature.

VGluT1 and Syp1 proteins are present in high copy numbers on synaptic vesicles (~9 and~31 per vesicle, respectively) (*Takamori et al., 2006*). As discussed above, it is possible that they are not completely retrieved after fusion (*Gimber et al., 2015*); and also see *Figure 4D–H*). In our experiments, about 40% of all vGluT1-pHluorin events and 50% of Syp1-pHTomato events do not show protein retrieval during the recording period, probably reflecting fully collapsed vesicles after fusion (ultraslow retrieval events). Taken together, these observations suggest that three modes of vesicle recycling take place; a mode where synaptic vesicle proteins remain at the surface membrane for extended time period albeit in a clustered fashion, an ultrafast mode of retrieval that occurs in the timescale of 150–300 ms, and a fast mode of endocytosis happening in the order of several seconds. To evaluate this premise further, we designed a probe with V0a1 subunit of the vacuolar ATPase (v-ATPase) responsible for vesicle acidification with pHluorin attached to one of its intraluminal loops (V0a1-pHluorin, *Figure 6*). V-ATPase and its subunits show low levels of expression in synapses and they have a low copy number per synaptic vesicle (~1) (*Takamori et al., 2006*). Co-localization experiments revealed that V0a1-pHluorin is properly trafficked to synapses, with about 10–30% of the probe co-localizing with the presynaptic marker Synapsin-1, also showing a similar subcellular distribution to the endogenous V0a1 protein (*Figure 6A–B*) and similar to what was described previously (*Bagh et al., 2017*). V0a1 over-expression or V0a1-pHluorin expression did not produce significant alterations in synaptic transmission, as indicated by similar to control single AP evoked response amplitudes and charge transfer, as well as similar paired-pulse ratios (*Figure 6—figure supplement 1A–D*). V0a1-pHluorin expression also had no effect on spontaneous, miniature excitatory postsynaptic current (mEPSC) amplitudes and frequency, and only caused a 2-fold increase in miniature inhibitory postsynaptic currents (mIPSC) frequency, with no changes in amplitude (*Figure 6C–E*). Application of a depolarizing high potassium solution accentuated synaptic localization and demonstrated activity-dependent trafficking of V0a1-pHluorin (*Figure 6F*). We used acid quenching (pH ~5) to suppress surface fluorescence followed by $NH_4Cl$ perfusion to alkalinize intracellular compartments and determined the ratio of intracellular and surface V0a1-pHluorin (*Figure 6G–H*). The subcellular distribution was highly variable among synapses, with ~40% of the probe present in internal organelles and ~60% on the plasma membrane on average, resembling

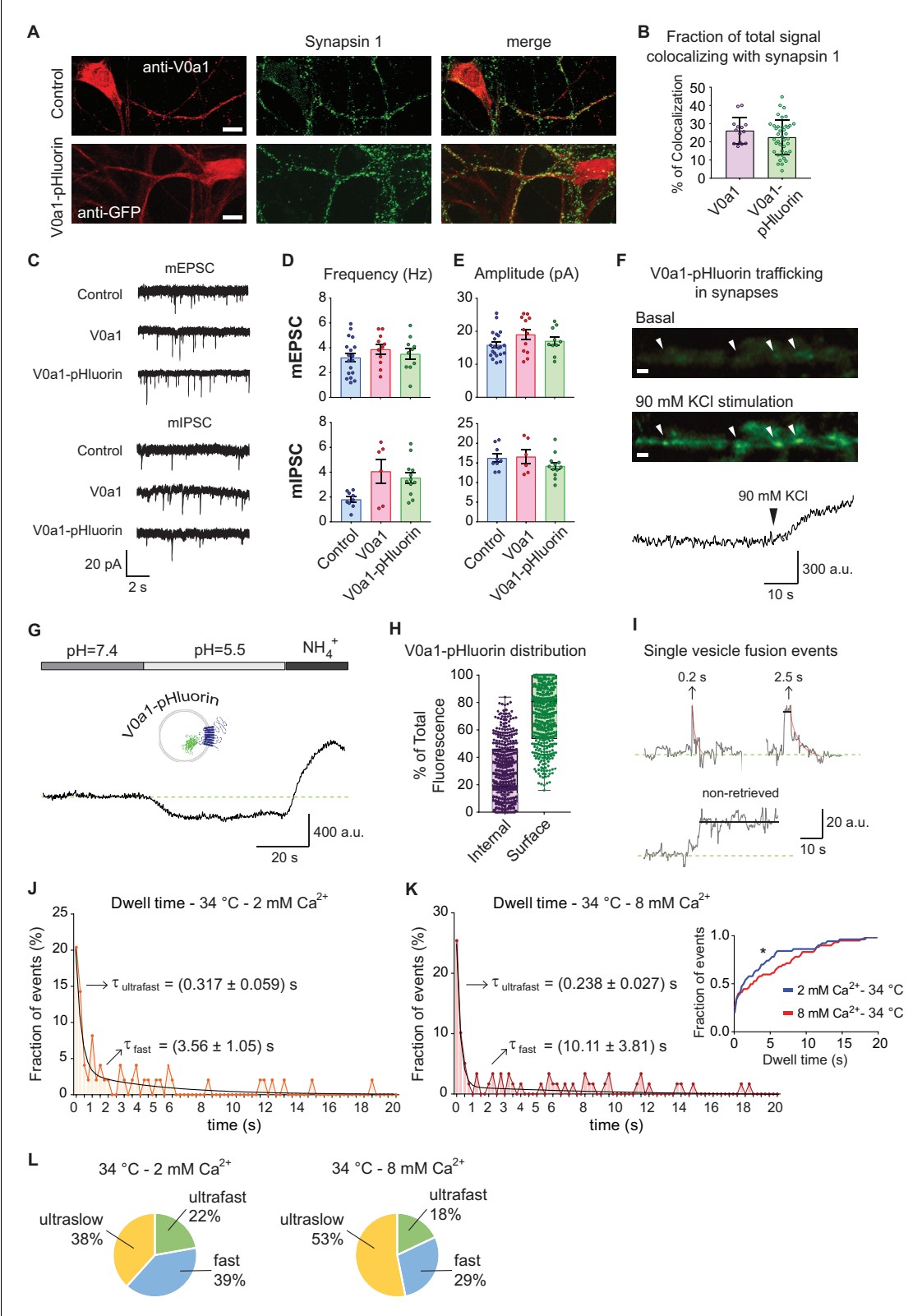

**Figure 6.** V0a1-pHluorin characterization and presynaptic trafficking. (**A**) Immunofluorescence of control (empty vector) and V0a1-pHluorin expressing hippocampal neurons. Top: control neuron stained with anti-V0a1 antibody to reveal endogenous V0a1 levels and distribution in RED. Bottom: V0a1-pHluorin expressing neuron immunostained with anti-GFP to show levels and distribution of the fusion protein in RED. For both, in GREEN staining against synapsin one is shown. White scale bar = 10 μm. (**B**) Quantification of colocalizing signal between Synapsin one and V0a1 or V0a1-pluorin.
*Figure 6 continued on next page*

*Figure 6 continued*

Colocalization analysis was object based, using a custom macro for Fiji. Positive colocalization was defined as an overlap in the area with above threshold signal in both channels after image segmentation. The % of colocalizing area was calculated. (C) Example traces of miniature inhibitory (mIPSC) and excitatory (mEPSC) postsynaptic currents for control (mock virus transfected), V0a1 over-expression and V0a1-pHluorin groups. Current clamp experiments were performed at room temperature and recorded for 5 min. (D) Frequency (Hz) of mIPSC (top) and mEPSC (bottom) for control, V0a1 and V0a1-pHluorin groups. (E) Amplitude (pA) of mIPSC (top) and mEPSC (bottom) for control, V0a1 and V0a1-pHluorin groups. For D to F: mIPSC: control = 9 cells; V0a1 = 7 cells; V0a1-pHluorin = 13 cells; mEPSC: control = control = 18 cells; V0a1 = 12 cells; V0a1-pHluorin = 10 cells. 1–3 neurons were patched per coverslip, 4–5 independent experiments (cultures). Also see the analysis of evoked neurotransmission in V0a1 and V0a1-pHluorin groups in *Figure 6—figure supplement 1*. (F) Representative wide-field fluorescence image from V0a1-pHluorin expressing neurons, before (basal signal) and after stimulation with 90 mM KCl, showing exocytosis of the probe. White arrowheads: presynaptic boutons. The quantification of fluorescence intensity over time for one of the boutons is shown in below images. White scale bars = 2 μm. (G) Example trace (average from one experiment) of V0a1-pHluorin fluorescence changes after Tyrode's buffer pH = 5.5 perfusion and $NH_4^+$ 50 mM application. (H) Quantification of the distribution (ratio) of V0a1-pHluorin in internal membranes (internal – purple) and plasma membrane (surface – green). 420 boutons analyzed from five coverslips (three independent experiments – cultures). (I) Example traces of de-noised single synaptic vesicle fusion events measured with V0a1-pHluorin, showing different dwell time lengths. Example traces after strong stimulation (40 Hz, 5 s) and folimycin treatment are shown in *Figure 6—figure supplement 1*. (J) Distribution of dwell time durations at 34°C and 2 mM extracellular $Ca^{2+}$. Black line: double exponential decay fit (R-square = 0.837; RRS = 0.01181). Arrows: decay constants for the fast and slow component of the exponential. 95 boutons from six coverslips. (K) Distribution of dwell time durations at 34°C and 8 mM extracellular $Ca^{2+}$. Black line: double exponential decay fit (R-square = 0.885; RRS = 0.009151). Arrows: decay constants for the fast and slow component of the exponential. (L) Pie charts depicting the ratio of each mode of retrieval respect to the total number of measured events, at 34°C and 2 or 8 mM extracellular $Ca^{2+}$. Green: ultrafast retrieval (dwell time duration between 0 and 1 s). Blue: fast endocytosis (dwell time of 1 to 20 s). Yellow: ultra-slow retrieval (>20 s). For J to L. 140 boutons from six coverslips. Inset: cumulative histogram comparing the effect of different $Ca^{2+}$ concentration at 34°C. Kolmogorov-Smirnov test of cumulative histogram: 34°C 2 mM $Ca^{2+}$ vs. 34°C 8 mM $Ca^{2+}$: p=0.0086.

DOI: https://doi.org/10.7554/eLife.36097.009

The following figure supplement is available for figure 6:

**Figure supplement 1.** Neurotransmission is not altered by V0a1 or V0a1-pHluorin overexpression.

DOI: https://doi.org/10.7554/eLife.36097.010

what was described for endogenous V0a1 (*Morel et al., 2003*). Due to the low copy number per synapse, 40 Hz stimulation only led to a small increase in fluorescence, which could be clearly visualized after treatment with folimycin to prevent re-acidification (*Figure 6—figure supplement 1E*). Nevertheless, we could detect single vesicle fusion events in response to single AP stimulation albeit with a low probability where ~ 50% of the ROIs did not respond to stimulation. Application of v-ATPase inhibitor folimycin (200 μM) converted 80–90% of these events to a non-decaying time course with a monophasic distribution of amplitudes supporting their basis in quantal fusion of single vesicles (*Leitz and Kavalali, 2011*; *Li et al., 2017*) (*Figure 6—figure supplement 1F–G*). The distribution of V0a1-pHluorin dwell times at 34°C could be fitted with a double exponential decay, again reflecting ultrafast and fast time constants of ~250–300 ms and 3–10 s, respectively (*Figure 6J–K*). As observed with the previous indicators, the time scale of the fast component of endocytosis was more sensitive to changes in extracellular $Ca^{2+}$ concentration, showing more than two-fold decrease in speed compared to a ~ 30% slow down of the ultrafast component. The proportion of each mode of endocytosis for V0a1-pHluorin retrieval is shown in *Figure 6L*.

## Detection of synaptic vesicle retrieval at rapid image acquisition settings

Our measurements suggest that the ultrafast endocytic mechanism occurs at an average speed of ~150–300 ms, consistent with the recent flash-and-freeze electron microscopy-based estimates of ultrafast endocytosis in hippocampal synapses (*Watanabe et al., 2013*). Although, these estimates are close to the Nyquist limit of detection based on our imaging speed of 10 Hz, our analysis of simulated single event traces (see *Figure 2—figure supplement 1C–E*) suggests that aliasing at this rate is not a major factor altering our measurements. Nevertheless, to address this potential concern, we repeated the experiments using vGluT1-pHluorin at an imaging speed of 40 Hz (*Figure 7*), taking advantage of the considerable improvement in signal-to-noise ratio and sensitivity added by signal de-noising (see negative controls and artificial traces test in *Figure 7—figure supplement 1B–I*). Example single synaptic vesicle fusion events are shown in *Figure 7A*. The distribution of dwell times calculated at 40 Hz imaging speed closely resembled those obtained at 10 Hz, with a peak at 200

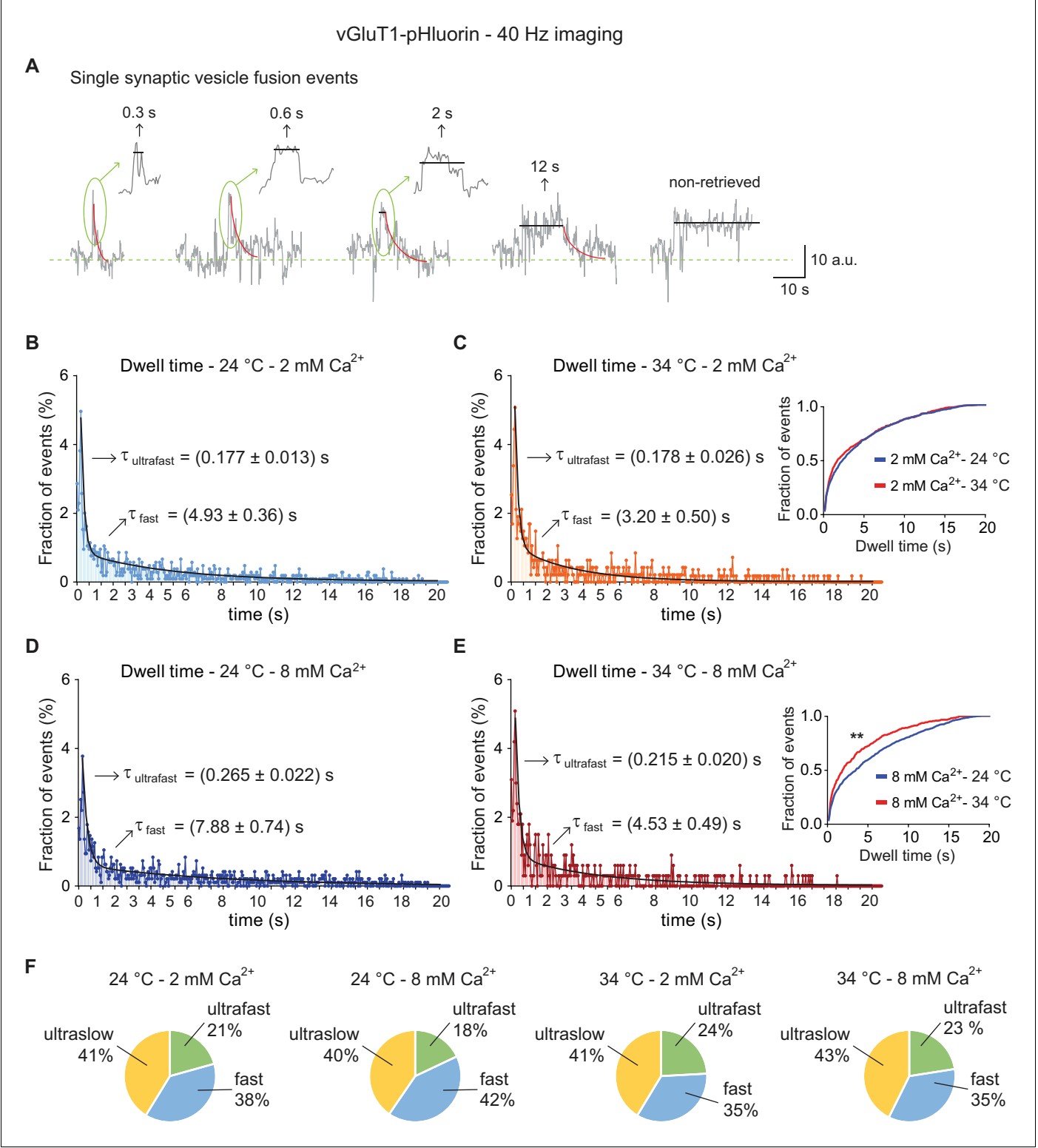

**Figure 7.** High-speed imaging of vGluT1-pHluorin corroborates the presence of ultrafast retrieval for single synaptic vesicle fusion events. (**A**) Example traces for single vesicle fusion events after de-noising, measured with vGluT1-pHluorin imaging at a speed of 40 Hz. Arrows and insets: expansion of the peak showing dwell time lengths in detail. (**B**) Distribution of dwell time durations at 24°C and 2 mM extracellular $Ca^{2+}$. Black line: double exponential decay fit (R-square = 0.8377; RRS = 0.00115). Arrows: decay constants for the ultrafast and fast component of the exponential. (**C**) Distribution of dwell time durations at 34°C and 2 mM extracellular $Ca^{2+}$. Black line: double exponential decay fit (R-square = 0.7768; RRS = 0.00121). Arrows: decay constants for the ultrafast and fast component of the exponential. Inset: cumulative histogram comparing the effect of temperature on

*Figure 7 continued on next page*

*Figure 7 continued*

dwell times at 2 mM Ca$^{2+}$, there is no significant effect. (**D**) Distribution of dwell time durations at 24°C and 8 mM extracellular Ca$^{2+}$. Black line: double exponential decay fit (R-square = 0.8667; RRS = 0.00119). Arrows: decay constants for the ultrafast and fast component of the exponential. (**E**) Distribution of dwell time durations at 34°C and 8 mM extracellular Ca$^{2+}$. Black line: double exponential decay fit (R-square = 0.770; RRS = 0.0023). Arrows: decay constants for the ultrafast and fast component of the exponential. Inset: cumulative histogram comparing the effect of temperature on dwell times at 8 mM Ca$^{2+}$, **=p < 0.01. (**F**) Pie charts depicting the relative contribution of the three modes of retrieval to total endocytosis for all experimental groups. Green: ultrafast retrieval (dwell time duration between 0 and 1 s). Blue: fast endocytosis (dwell time of 1 to 20 s). Yellow: ultra-slow retrieval (>20 s). Note that the percentage of each type of endocytosis is not greatly affected by changes in Ca$^{2+}$ or temperature. For B to E. Kolmogorov-Smirnov test: 24°C 2 mM Ca$^{2+}$ vs. 24°C 8 mM Ca$^{2+}$: p<0.0001; 34°C 2 mM Ca$^{2+}$ vs. 34°C 8 mM Ca$^{2+}$: p=0.0043; 24°C 2 mM Ca$^{2+}$ vs 34°C 2 mM Ca$^{2+}$: p=0.08271; 24°C 8 mM Ca$^{2+}$ vs 34°C 8 mM Ca$^{2+}$: p<0.0001. Kruskal-Wallis test: p<0.0001; Dunn's post-test: 24°C 2 mM Ca$^{2+}$ vs. 24°C 8 mM Ca$^{2+}$: p<0.0001; 34°C 2 mM Ca$^{2+}$ vs. 34°C 8 mM Ca$^{2+}$: non-significant; 24°C 2 mM Ca$^{2+}$ vs 34°C 2 mM Ca$^{2+}$: non-significant; 24°C 8 mM Ca$^{2+}$ vs 34°C 8 mM Ca$^{2+}$: p<0.0001. For all the data presented in this figure: 24°C – 2 mM Ca$^{2+}$: 558 boutons from eight coverslips; 24°C – 8 mM Ca$^{2+}$: 474 boutons from seven coverslips; 34°C – 2 mM Ca$^{2+}$: 327 boutons from seven coverslips; 34°C – 8 mM Ca$^{2+}$: 209 boutons from five coverslips. At least three independent experiments (cultures). Also see *Figure 7—figure supplement 1* for analysis of negative controls and simulated traces. Legends to the figure supplements.

DOI: https://doi.org/10.7554/eLife.36097.011

The following figure supplement is available for figure 7:

**Figure supplement 1.** False positive artifacts are negligible at fast (40 Hz) imaging speeds.

DOI: https://doi.org/10.7554/eLife.36097.012

ms and a double exponential behavior (*Figure 7B–E*). The mean endocytic speed for the ultrafast component of the distribution is ~150–250 ms, 3–5 times higher than the Nyquist limit. As observed before, at both 24°C and 34°C, endocytosis is slowed down by higher Ca$^{2+}$ concentration, although this effect is more pronounced at room temperature and particularly for the fast endocytic component (at 24°C there is a ~50 and ~60% increase in decay time at 8 mM Ca$^{2+}$ for the ultrafast and fast components, respectively, while at 34°C the change is ~20 and ~40%, respectively; see arrows in *Figure 7B–E*). Statistical analysis of the distributions corroborated this observation, demonstrating that the overall effect of Ca$^{2+}$ rise in the whole event distribution is only significant at 24°C while the effect of temperature rise is significant exclusively at 8 mM extracellular Ca$^{2+}$ concentration (insets in *Figure 7D and E*, and legend). As observed previously, temperature and Ca$^{2+}$ did not have a major impact in the proportion of each mode of endocytosis (*Figure 7F*).

## Discussion

In this study, using a combination of fluorescent reporters based on the vesicular glutamate transporter, synaptophysin or the V0a1 subunit of the vacuolar ATPase, we were able to identify three kinetically different pathways that retrieve individual synaptic vesicles after single AP driven fusion. As the retrieval patterns of these three probes largely overlapped, we could draw conclusions not only on the kinetic pathways that govern the retrieval of individual proteins but also on retrieval of associated synaptic vesicles.

Visualization of vesicle retrieval was aided by rapid image acquisition settings and a time-domain signal de-noising algorithm adapted from earlier work focusing on estimation of ion channel kinetics parameters from single channel recordings (*Chung and Kennedy, 1991*). Furthermore, in these experiments monitoring endocytosis of single synaptic vesicles following fusion helped us to uncouple regulation of synaptic vesicle fusion by temperature and Ca$^{2+}$ from the putative effects of these factors on retrieval. Since synaptic vesicle fusion occurs in a binary fashion, once a vesicle is fused its subsequent trajectory can be kinetically isolated from its fusion probability. Therefore, the nature of single vesicle fusion eliminates the requirement to normalize the extent of endocytosis to exocytosis and thus enables isolation of potential direct effects on endocytosis.

One of the pathways we visualized retrieved synaptic vesicle proteins with a time scale on the order of 200 milliseconds. This time frame is consistent with ultrafast endocytosis as the time course is an order of magnitude faster than earlier estimates of fast endocytosis (*Klingauf et al., 1998*; *von Gersdorff and Matthews, 1994*) and in line with the time scales detected in recent EM-based or capacitance-based methods (*Delvendahl et al., 2016*; *Watanabe et al., 2013*). In addition to the ultrafast retrieval of fluorescent probes, we also detected, as in earlier optical experiments, a 'fast' mode of endocytosis operating within seconds as well as an ultraslow form of trafficking where we

could not reliably determine a decrease in signal after fusion for the duration of our optical recordings (*Gandhi and Stevens, 2003*; *Zhu et al., 2009*). Importantly, the parallel operation of the three pathways persisted in response to changes in temperature or $Ca^{2+}$ levels as we did not observe major shifts in the relative segregation and composition of the dwell time distributions. However, the intrinsic kinetics of the fast pathway were modulated by temperature and $Ca^{2+}$, whereas the intrinsic kinetics of the ultrafast pathway were relatively unperturbed by these manipulations.

As indicated above, our estimates for the ultrafast endocytosis are in line with previous measurements using EM based methods or capacitance measurements in multiple systems. For instance, experiments using 'flash-and-freeze' electron microscopy (*Watanabe et al., 2013*) revealed an ultrafast pathway of endocytosis that takes place at physiological temperature, but is not present or measurable at room temperature. Through this mechanism large invaginations appeared as soon as 50 ms after optogenetic stimulation and fully endocytosed membranes, endosome-like closed structures, peaked at 100–300 ms (*Watanabe et al., 2013*). Subsequent capacitance measurements in hippocampal and cerebellar neurons corroborated this temperature-dependent ultrafast kinetics, revealing a mean time constant of ~470 ms for single AP evoked endocytic events only at 36°C (*Delvendahl et al., 2016*).

However, previous capacitance measurements in pituitary terminals as well as at the calyx of Held, which were performed at room temperature, reported the presence of fast flickering fusion pores in neurons with a mean open duration of ~300 ms (*Klyachko and Jackson, 2002*; *He et al., 2006*). These fast flickering pores constituted around 20% of all exocytic events (*He et al., 2006*). In another report, the time constant of endocytosis after vesicle fusion was estimated to be in the order of 100 ms (*Sun et al., 2002*). These earlier findings would imply that ultrafast retrieval could occur at room temperature. In agreement with this premise, our experiments show that endocytosis of distinct synaptic vesicle proteins after single AP stimulation occur through similar kinetically distinguishable mechanisms. The ultrafast endocytosis process we measured has a mean duration of 150–250 ms, consistent with the time course of ultrafast endocytosis reported previously. Importantly, the fastest speed at which synaptic vesicle proteins can be fully retrieved after exocytosis seems to be ~150 ms, since the probability distribution of dwell times shows a dramatic reduction for shorter times. In our measurements, the mean speed of ultrafast endocytosis was only mildly affected either by changes in temperature or $Ca^{2+}$ concentration, suggesting that the underlying molecular mechanism is insensitive to these factors and probably already operating at the limiting speed. The vast majority of ultrafast endocytotic events in our experiments showed quantal retrieval — that is the same number of proteins that fused were retrieved — indicating that synaptic vesicle molecular identity can be conserved through ultrafast endocytosis. One possible mechanism underlying this observation could be the re-closure of the fusion pore (compatible with kiss-and-run) where proteins remain on the synaptic vesicle membrane and are unable to diffuse away to the plasma membrane (*Alabi and Tsien, 2013*). Additionally, the high efficiency in protein retrieval during ultrafast endocytosis could also be explained by a mechanism where synaptic vesicle components, lipids and proteins, remain clustered in the plasma membrane after fusion as was proposed previously (*Bennett et al., 1992*; *Willig et al., 2006*; *Opazo and Rizzoli, 2010*). Subsequently, these clusters could be rapidly retrieved following a mechanism similar to the one found using rapid freeze EM methods (*Watanabe et al., 2013*).

Our results showed that at the level of single vesicle retrieval, $Ca^{2+}$ appears to predominantly target fast events that occur in the order of seconds but leave the time course of ultrafast events relatively unperturbed. Recent work from our group showed that synaptotagmins, besides their role in coupling fusion to $Ca^{2+}$ signals, can also regulate the endocytic time course of single vesicle fusion events in a $Ca^{2+}$-dependent manner (*Leitz and Kavalali, 2016*; *Li et al., 2017*). Particularly, in the absence of the main $Ca^{2+}$-sensor for synchronous release, syt1, ~80% of endocytosis occurs only through a rapid retrieval mechanism – less than 1 s – with no detectable delays (also see *Li et al., 2017*). Moreover, in the absence of syt1 we detected very limited excess or partial retrieval of synaptic vesicle proteins. This finding suggests that the syt1-dependent delay in vesicle retrieval kinetics also to some extent disrupts the fidelity of vesicle protein retrieval, possibly facilitating the generation of vesicles with diverse protein compositions during retrieval (*Crawford and Kavalali, 2015*; *Raingo et al., 2012*). Furthermore, the kinetics of single vesicle protein retrieval are not altered by syt7 knock down and knocking down syt7 on top of syt1 KD does not further accelerates endocytosis (*Li et al., 2017*). Taken together, these results raise the possibility that ultrafast endocytosis is

relatively syt1 and Ca$^{2+}$-independent, while fast endocytosis might be regulated by these factors, supporting the notion that different molecular pathways govern the two processes. Previous reports of ultrafast endocytosis showed that it does not involve a clathrin-mediated pathway (*Delvendahl et al., 2016*; *Watanabe et al., 2013*), but it is dependent on actin. In this regard, a recent study implicated formins — actin remodeling proteins that serve several cell biological functions — in synaptic vesicle endocytosis in hippocampal synapses as well as at the calyx of Held terminals (*Soykan et al., 2017*). Moreover, GTP and dynamin independent endocytosis have been proposed to coexist in neurons with dynamin-dependent pathways (*Van Hook and Thoreson, 2012*; *Xu et al., 2008*). Taken together with our results, this indicates that a novel, still molecularly uncharacterized mechanism may operate independently of the classical endocytic machinery and synchronize synaptic vesicle retrieval with fusion in the timescale of milliseconds. As mentioned before, a tempting mechanism that fits this description is kiss-and-run, although further experimental support is needed to validate this hypothesis (*Alabi and Tsien, 2013*; *Chanaday and Kavalali, 2017*).

Around 40% of the fusion events detected in our experiments were not endocytosed during the imaging period (20 s), indicating that even for single synaptic vesicle exocytosis slow modes of retrieval contribute to synaptic vesicle recycling. Elegant studies showed that synaptic vesicle proteins undergo localized diffusion after exocytosis, followed by re-clustering and endocytosis (*Gimber et al., 2015*). The re-clustering and re-capturing of synaptic vesicle proteins is an extremely slow process, requiring more than 60 s, and it was mediated by the clathrin adaptors CALM and AP-180 (*Gimber et al., 2015*). The ultraslow retrieval presented here could be compatible with this mechanism.

The speed of endocytosis is a limiting step in the maintenance of synaptic transmission under repetitive stimulation (*Kavalali, 2006*). Synaptic vesicle endocytosis can rapidly replenish vesicle pools and prevent vesicle depletion, while removing the potential hindrance of subsequent fusion events by previously fused vesicles (*Fernández-Alfonso and Ryan, 2004*; *Hua et al., 2013*; *Sara et al., 2002*). Accordingly, it is plausible to envision the physiological necessity of an ultrafast endocytic mechanism clearing future sites of fusion in preparation for subsequent rounds of neurotransmitter release. We believe the single vesicle imaging approach we present here will facilitate the identification of heretofore poorly understood mechanisms underlying the ultrafast vesicle retrieval process.

## Materials and methods

**Key resources table**

| Reagent type | Designation | Source or reference | Identifiers | Additional information |
|---|---|---|---|---|
| Antibody | Mouse monoclonal anti-GFP | Cell Signaling | Catalog # 29565 | dil. 1:100 |
| Antibody | Rabbit polyclonal anti-V0a1 | Synaptic Systems | Catalog # 109 002 | dil. 1:100 |
| Antibody | Mouse monoclonal anti-Synapsin 1 | EMD Millipore | Catalog # MABN894 | dil. 1:1000 |
| Antibody | Mouse monoclonal anti-GDI | Synaptic Systems | Catalog # 130 011 | dil. 1:5000 |
| Chemical, compound, drug | 6-Cyano-7-nitroquinoxaline -2,3-dione disodium salt hydrate (CNQX) | Sigma-Aldrich | Catalog # C239 | 10 µM |
| Chemical, compound, drug | D(−)−2-Amino-5-phosphonopentanoic acid (AP-5) | Sigma-Aldrich | Catalog # A8054 | 50 µM |
| Chemical, compound, drug | Tetrodotoxin (TTX) | Enzo Life Sciences | Catalog # BML-NA120-0001 | 1 µM |
| Chemical, compound, drug | Picrotoxin (PTX) | Sigma-Aldrich | Catalog # P1675 | 50 µM |

*Continued on next page*

*Continued*

| Reagent type | Designation | Source or reference | Identifiers | Additional information |
|---|---|---|---|---|
| Chemical, compound, drug | Folimycin from Streptomyces sp. | Calbiochem/EMD | Catalog # 344085 | 200 nM |
| Chemical, compound, drug | Trypsin from bovine pancreas | Sigma-Aldrich | Catalog # T9935 | |
| Chemical, compound, drug | DNase | Sigma-Aldrich | Catalog # D5025-375KU | |
| Chemical, compound, drug | Matrigel | Corning | Catalog # 354234 | dil. 1:25 |
| Chemical, compound, drug | FuGENE 6 | Promega | Catalog # E2692 | |
| Chemical, compound, drug | QX-314 | EMD-Millipore | Catalog # 552233 | |
| Cell line | Highly transfectable derivative of human embryonic kidney-293 epithelial adherent cells (HEK293T) | ATCC | Catalog # CRL-1573 | |
| Organism/Strain | Sprague-Dawley rat pups (P2-P4) | | | |
| Recombi-nant DNA | Plasmid: pRSV-REV (lentiviral packaging) | Addgene | Catalog # 12253 | |
| Recombi-nant DNA | Plasmid: pCMV-VSV-G (lentiviral packaging) | Addgene | Catalog # 8454 | |
| Recombi-nant DNA | Plasmid: pMDLg/pRRE (lentiviral packaging) | Addgene | Catalog 12251 | |
| Recombi-nant DNA | Plasmid: pFUGW-vGlut1 -pHGFP | *Voglmaier et al. (2006)* | N/A | |
| Recombi-nant DNA | Plasmid: pFU-Syp1 -pHTomato | *Li and Tsien (2012) Leitz and Kavalali (2011)* | N/A | |
| Recombi-nant DNA | Plasmid: pFUGW-Va01 | (see Materials and methods section) | | |
| Recombi-nant DNA | Plasmid: pFUGW-Va01 -pHluorin | (see Materials and methods section) | | |
| Software and Algorithms | Forward-backward non-linear filter: noise reduction (NoiseReduc) function for Matlab from Nigel Reuel | *Chung and Kennedy (1991) Reuel et al. (2012)* | http://web.mit.edu/stranogroup/ index.php/resources/19-simulation-and- analysis-codes/norse-code-files/40-norse- algorithm.html | |
| Software and Algorithms | Single-vesicle-fusion-events | Present work | https://github.com/nchanaday/ Single-vesicle-fusion-events | Copy archived at https://github. |

## Dissociated hippocampal cultures

Postnatal day 2–4 Sprague-Dawley rats were used for the experiments. Both hippocampi were dissected in sterile conditions and posteriorly dissociated using 10 mg/ml trypsin and 0.5 mg/ml DNAase for 10 at 37°C. After careful trituration using a P1000 pipette, cells were resuspended to a concentration of 1 pups per 16 coverslips and plated onto 12 mm coverslip coated with 1:25 MEM: Matrigel solution. Basic growth medium consisted of MEM medium (no phenol red), 5 g/l D-glucose, 0.2 g/l $NaHCO_3$, 100 mg/l transferrin, 5% of heat inactivated fetal bovine serum, 0.5 mM L-glutamine, 2% B-27 supplement, and 2–4 µM cytosine arabinoside. Cultures were kept in humidified incubators at 37°C and gassed with 95% air and 5% $CO_2$.

## Cloning

The super-ecliptic pHluorin was inserted between Gly-677 and Thr-678 of V0a1 from Mus musculus (Atp6v0a1 gene – UniProtKB database number Q9Z1G4). The construct was subcloned into pFU-GW lentiviral vector from Addgene.

## Lentiviral infection

Lentiviruses were produced in HEK293T cells (catalog number CRL-1573; ATCC, Manassas, VA, US) by contransfection of pFUGW transfer vectors and three packaging plasmids (pCMV-VSV-G, pMDLg/pRRE, pRSV-Rev) using Fugene six transfection reagent (catalog number E2692; Promega, Madison, WI, US). The supernatants of the cultures were collected 72 hr after the transfection and clarified by centrifugation (2000 rpm 15 min), and subsequently used for infection of DIV four hippocampal neurons. All experiments were performed on 16–20 DIV cultures when synapses were mature and lentiviral expression of constructs of interest was optimal (*Mozhayeva et al., 2002*; *Deák et al., 2006*). All experiments were performed following protocols approved by the UT Southwestern Institutional Animal Care and Use Committee.

## Western blotting

Western blots were performed as described in Nosyreva and Kavalali (2010). Primary antibodies against GDI and V0a1 were used in 1:1000 and 1:500 dilution, respectively. Immunoreactive bands were visualized by enhanced chemiluminescence (ECL), captured on autoradiography film and analyzed using GelAnalyzer2010 software (http://www.gelanalyzer.com). V0a1 protein levels were normalized to GDI loading control.

## Immunofluorescence

16–18 DIV neuron cultures were fixed for 10 min in PBS containing 4% para-formaldehyde (PFA) and processed for as previously described (*Ramirez et al., 2008*). Primary antibody against GFP was used to detect pHluorin-tagged proteins (1:200), antibody against V0a1 subunit of the V-ATPase was used to detect total V0a1 in control and V0a1-pHluorin expressing neurons (1:250), and anti-Synapsin1 antibody was used to detect presynaptic boutons (1:1000; control experiments were performed using Synaptobrevin two antibody to corroborate the results). Alexa-conjugated secondary antibodies (1:1000) were used to label the cells and then coverslips were mounted and imaged using an LSM 510 META confocal microscope (Carl Zeiss, Oberkochen, Germany) with a 63X (NA1.4) objective.

## Electrophysiology

Cultured pyramidal neurons between 14 to 18 DIV were used for whole cell recordings at a clamped voltage of −70 mV by means of Axopatch 200B and Clampex 8.0 software (Molecular Devices, San Jose, CA, US), filtering at 2 kHz and sampling at 5 kHz. The cells were visualized using a Nikon DIAPHOT 200 microscope (Nikon, Minato, Tokyo, Japan). The internal pipette solution contained 115 mM $CsMeSO_3$, 10 mM CsCl, 5 m M NaCl, 10 mM HEPES, 0.6 mM EGTA, 20 mM tetraethylammonium chloride, 4 mM Mg-ATP, 0.3 mM $Na_2GTP$ and 10 mM QX-314 (lidocaine N-ethyl bromide). The final solution was adjusted to pH 7.3 and 304 mOsM. Final resistance of the electrode tips was ~3–6 MΩ. For all experiments, the extracellular solution was a modified Tyrode's solution containing 150 mM NaCl, 4 mM KCl, 10 mM glucose, 10 mM HEPES, 2 mM $MgCl_2$ and 2 mM $CaCl_2$, adjusted to pH 7.4 and 310 mOsM. To isolate inhibitory postsynaptic currents, agonists of ionotropic glutamate receptors were added: 10 µM 6-cyano-7-nitroquinoxaline-2,3-dione (CNQX) and 50 µM aminophosphonopentanoic acid (AP-5). To isolate excitatory currents (AMPA-mediated) 50 µM AP-5 and 50 µM picrotoxin (PTX, ionotropic GABA receptor inhibitor) were added to the bath solution. To elicit evoked responses, electrical stimulation was delivered through parallel platinum electrodes with a constant current unit (WPI A385; World Precision Instruments, Sarasota, FL, US) set at 35 mA. Spontaneous activities (mIPSCs and mEPSCs) were recorded with the addition of 1 µM TTX. Miniature events were identified with a 5 pA detection threshold and analyzed with MiniAnalysis (Synaptosoft, Fort Lee, NJ, US).

## Fluorescence imaging

Cultured hippocampal neurons at 16–20 DIV transfected with either vGluT1-pHluorin, Syp1-pHTomato or V0a1-pHluorin were used for the imaging experiments. The modified Tyrode's buffer from above containing 2 or 8 mM $Ca^{2+}$ was used with 10 µM CNQX and 50 µM AP-5 to prevent recurrent network activity. For experiments performed at ~34°C solutions were heated using a bipolar temperature controller (CL-1000) attached to a multi-line solution heater (SHM-828; Warner Instruments,

Hamden, CT). The objective was heated with an objective collar connected to a single channel temperature controller (H401-T-SINGLE-BL; Okolab, Shanghai, China), and the stage and microscope were isolated from the room with a protective case to minimize temperature variances. Fluorescence was recorded using a Nikon Eclipse TE2000-U microscope with a 100X Plan Fluor objective (Nikon, Minato, Tokyo, Japan) attached to an Andor iXon + back illuminated EMCCD camera (Model no. DU-897E-CSO-#BV; Andor Technology, Belfast, UK). For illumination, we used a Lambda-DG4 illumination system (Sutter Instruments, Novato, CA, US) with a FITC or TRITC filter. Images were acquired at 10 or 40 Hz with binning of 4 by four to optimize the signal-to-noise ratio. Neurons were stimulated using parallel bipolar electrodes (FHC, Bowdoin, ME, US) delivering 35 mA pulses at 20 s intervals, followed by a rest period prior to the delivery of 200 APs at 40 Hz. Boutons were visualized by the addition of Tyrode's solution with 50 mM $NH_4Cl$ at the end of each experiment. Circular regions of interests (ROIs) of 2.27 μm diameter were automatically drawn around local fluorescence maximums using a custom-made macro for Fiji (*Schindelin et al., 2012*) and the fluorescent traces obtained were exported to Matlab (Mathworks, Natick, MA, US) for analysis.

To calculate surface versus intracellular distribution of V0a1-pHluorin, we perfused a modified Tyrode's solution at pH = 5.5 (buffered by MES instead of HEPES) to quench surface pHluorin and recorded for 30 s. After imaging putative boutons for another 30 s, a modified Tyrode's solution containing 50 mM $NH_4Cl$ was perfused for 30 s, in order to alkalinize all compartments. The difference of the mean fluorescence during acid buffer perfusion and normal Tyrode's solution correspond to the surface pool of pHluorin, while the difference in mean fluorescence between $NH_4Cl$ perfusion and normal Tyrode's solution correspond to the internal pool of pHluorin.

## Fluorescence analysis

Fluorescence intensity traces were analyzed using a custom made Matlab script (*Chanaday, 2018*; https://github.com/nchanaday/Single-vesicle-fusion-events; copy archived at https://github.com/elifesciences-publications/Single-vesicle-fusion-events), based on our previous analysis with some modifications (*Leitz and Kavalali, 2011*). Photobleaching was corrected with a single exponential decay and background was subtracted linearly, both photobleaching and background values were calculated based on fluorescence measurements of background in each imaging experiment and also for each ROI. De-noising was performed using a time-domain forward-backward non-linear filter developed originally by *Chung and Kennedy (1991)* and implemented for Matlab by Nigel Reuel (*Reuel et al., 2012*). Minor changes were made to the code in order to improve de-noising of our data, based on the original paper by *Chung and Kennedy (1991)*. To find single vesicle fusion events, successful events were defined as those whose fluorescence amplitude was greater than three times the standard deviation of the baseline (average of ~2 s prior to the event). To avoid multivesicular fusion events, the upper limit was set at the mean value of single event amplitude plus half the difference between the mean single vesicle amplitude and the mean of the next amplitude peak in the distribution (corresponding to two quantums). This value was calculated comparing amplitude distributions for experiments performed in different extracellular $Ca^{2+}$ concentration with or without TTX and with or without folymicin. Also, the event time has to be coincident with the time of stimulation. Dwell times were calculated as the time between the initial fluorescence step and the start of fluorescence decay defined as a switch to negative values of the first derivative. For 40 Hz stimulation, amplitude measurement, single exponential decay fitting and rise slope linear fitting were also performed in an automatized way using Matlab (Mathworks, Natick, MA, US).

The authors are open to share the Matlab script developed by our lab for the analysis of single synaptic vesicle fusion and endocytosis events used in the present work, upon request to ETK or NLC.

## Statistical analysis

Histograms of single vesicle dwell time distributions were fitted using Matlab and OriginPro 8.1 (OriginLab, Northampton, MA, US), to corroborate the results. The comparison of reduced R-square and F (from F test) values between single exponential decay and double exponential decay fittings revealed a better fit (higher R-square and F values) for the double exponential model. R-square and residuals are informed in the figure legends.

N for each group and experiment are informed in the figure legends. The Kolmogorov–Smirnov (K-S) test was used to determine differences in cumulative probability histograms when comparing two groups, for three or more groups histograms were compared using Kruskal-Wallis analysis of medians and Dunn's multiple comparison post-test. Averaged fluorescence traces are shown as mean ± SD. Bar graphs always inform mean values ± SEM, except for non-parametric data where they express median ±confidence interval (5–95%).

## Acknowledgements

We are grateful to Brent Trauterman and Dr Robin Hiesinger for their assistance in designing and cloning V0a1-pHluorin. We would like to thank Ying Li, Patricia Horvath and Drs Helmut Kramer, Denise Ramirez and Kanzo Suzuki for their valuable discussion and comments on the manuscript. This work was supported by a grant from the National Institute of Mental Health (MH066198).

## Additional information

### Funding

| Funder | Grant reference number | Author |
| --- | --- | --- |
| National Institute of Mental Health | MH066198 | Ege T Kavalali |

The funders had no role in study design, data collection and interpretation, or the decision to submit the work for publication.

### Author contributions

Natali L Chanaday, Conceptualization, Software, Formal analysis, Validation, Investigation, Visualization, Methodology, Writing—original draft, Project administration, Writing—review and editing; Ege T Kavalali, Conceptualization, Resources, Supervision, Funding acquisition, Validation, Visualization, Writing—original draft, Project administration, Writing—review and editing

### Author ORCIDs

Natali L Chanaday https://orcid.org/0000-0002-3376-5187
Ege T Kavalali https://orcid.org/0000-0003-1777-227X

### Ethics

Animal experimentation: All animal protocols were approved by the Institutional Care and Use Committee at UT Southwestern Medical Center. The work presented in this study is covered by the Animal Protocol Numbers APN 2016-101416.

### Decision letter and Author response

Decision letter https://doi.org/10.7554/eLife.36097.016
Author response https://doi.org/10.7554/eLife.36097.017

## Additional files

### Supplementary files

• Transparent reporting form
DOI: https://doi.org/10.7554/eLife.36097.013

### Data availability

All data generated or analysed during this study are included in the manuscript and supporting files as histograms or box plots.

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
