## [Decision Letter]

Thank you for submitting your article "Optical detection of three modes of endocytosis at hippocampal synapses" for consideration by *eLife*. Your article has been reviewed by 3 peer reviewers, and the evaluation has been overseen by a Reviewing Editor and Richard Aldrich as the Senior Editor. One of the reviewers, Vitaly Klyachko, has agreed to reveal his identity.

The reviewers have discussed the reviews with one another and the Reviewing Editor has drafted this decision to help you prepare a revised submission.

Summary:

This exciting study addresses a long-standing controversy about the prevalent forms of vesicle retrieval in central synapses. Single vesicle endocytosis at hippocampal synapses is monitored with high time resolution using the vesicular glutamate transporter, synaptophysin, and the V0a1 subunit of the vacuolar ATPase as probes. A computational approach is used to resolve individual endocytic events with a very high temporal resolution. The study reveals different kinetically distinct pathways for retrieval of synaptic vesicles. It goes beyond previous studies since it measures the relative percentages of the various retrieval processes. Ultrafast events (150-250) are in the minority (20%), while 40% of the events proceed on a 5-12 sec timescale, and the remaining 40% on an even slower timescale. Different extracellular calcium concentrations and temperatures primarily influence the fast and slow retrieval processes.

Essential revisions:

Figure 4: The finding that Syt1 KD prefers the ultrafast pathway is remarkable. Please speculate if another calcium sensor (e.g., Syt7) might take over, and if the data are consistent with such a scenario. Or, are the data more consistent with an entirely different calcium sensor or a calcium independent process?

At high extracellular calcium concentration, the rate of "fast" and "slow" endocytosis is slowed down (but not that of ultrafast endocytosis). Could this be in part also caused by higher synaptic vesicle exocytosis activity, i.e., the balance between fusion events and retrieval is shifted at higher extracellular calcium concentration?

Figure 2—figure supplement 1. An event-by-event comparison of simulated vs. recovered events in addition to the cumulative distributions could offer a better assessment of the reliability of the computation approach used.

Figure 3. It would be informative to add a comparison of the event amplitudes for different forms of retrieval.

Discussion: Re-acidification kinetics at least in synaptosomes can be on the order of 500 ms (Budzinski et al. ACS Chem Neurosci, 2011). Please provide a brief discussion of the implications of the reacidification kinetics to the apparent time course of ultra-fast events.

Since the different internalization pathways may shuttle pHluorin molecules into different sized structures, possibly with different numbers of H-ATPases, which could be regulated differentially (e.g. due to different phospholipid content), it seems plausible that acidification rates may vary between the different routes of internalization. The approach used here is one of the few ways that one could look at this by looking at the exponential time constant of the decay in fluorescence. It would be interesting to see if there is a difference in these rates depending on dwell time. For example, in Figure 3A, the fast events (0.1s, 0.3s, 2.4s) appear to acidify faster than the slow events. If it holds for the larger data set, then this would be a new finding.

The authors state that overexpression of V0a1-pHluorin results in a "small" increase in mIPSC frequency. Actually, the figure shows a 2-fold increase. Also, does overexpression have an effect on evoked IPSCs?

The authors suggest that the results in Figure 6L differ from those in Figure 3. Please provide statistics to assess if the differences are significant.

---

## [Author Response]

Essential revisions:

Figure 4: The finding that Syt1 KD prefers the ultrafast pathway is remarkable. Please speculate if another calcium sensor (e.g., Syt7) might take over, and if the data are consistent with such a scenario. Or, are the data more consistent with an entirely different calcium sensor or a calcium independent process?

This is a very interesting question. Based on previous findings from our lab (Li et al., 2017), the kinetics of single vesicle endocytosis events are not altered by knocking down syt7. Moreover, the double knock-down of syt1 and syt7 showed the same fast endocytic phenotype as syt1 KD alone (Figure 6H-I in Li et al., 2017). These published results are also supported by averaged single vesicle traces of syt7 deficient synapses where we detected a double exponential decay (τ_1_= 0.31 ± 0.05 s, τ_2_= 8.5 ± 4.8 s). This result is consistent with two retrieval processes operating in the absence of syt7, similar to controls. Taken together, these findings suggest that the ultrafast mechanism detected by our experiments is syt7 independent and also possibly Ca^2+^-independent, while the fast pathway relies exclusively on syt1 for Ca^2+^ sensing.

We added this information about syt7 to the Results (subsection “Monitoring endocytosis with vGluT1-pHluorin”, third paragraph) and the Discussion (6th paragraph) sections.

At high extracellular calcium concentration, the rate of "fast" and "slow" endocytosis is slowed down (but not that of ultrafast endocytosis). Could this be in part also caused by higher synaptic vesicle exocytosis activity, i.e., the balance between fusion events and retrieval is shifted at higher extracellular calcium concentration?

We agree with the reviewers, therefore, in this study, we restricted our analysis to single vesicle fusion events by discarding multi-vesicular events as marked by their multiquantal higher amplitudes (see also Leitz and Kavalali, 2011). Therefore, we do not think the slow-down in retrieval is due to a higher exocytotic load but rather to specific Ca^2+^ regulation of the machinery involved in endocytosis (see Materials and methods section).

Figure 2—figure supplement 1. An event-by-event comparison of simulated vs. recovered events in addition to the cumulative distributions could offer a better assessment of the reliability of the computation approach used.

We agree with this suggestion, we added an event-by-event comparison with the respective error for each event (i.e. differences in dwell time duration) to Figure 2—figure supplement 1.

Figure 3. It would be informative to add a comparison of the event amplitudes for different forms of retrieval.

The mean peak amplitude of the fusion events is similar for all the forms of retrieval we detected, strengthening the premise that single synaptic vesicle fusion is followed by kinetically different endocytic pathways (please see Author response image 1). We now reference this finding in the Results section of the revised manuscript (subsection “Monitoring endocytosis with vGluT1-pHluorin”, third paragraph).

Discussion: Re-acidification kinetics at least in synaptosomes can be on the order of 500 ms (Budzinski et al. ACS Chem Neurosci, 2011). Please provide a brief discussion of the implications of the reacidification kinetics to the apparent time course of ultra-fast events.

A major strength of our analysis is that we can dissociate dwell time measures from the subsequent decay time course of individual events. This point is bolstered by the new analysis we present in Author response image 1, demonstrating that there is no detectable correlation between our dwell time and decay time estimates from single events (Author response image 1). We now refer to this finding in the Results section of the manuscript (subsection “Monitoring endocytosis with vGlutT1-pHluorin”).

Since the different internalization pathways may shuttle pHluorin molecules into different sized structures, possibly with different numbers of H-ATPases, which could be regulated differentially (e.g. due to different phospholipid content), it seems plausible that acidification rates may vary between the different routes of internalization. The approach used here is one of the few ways that one could look at this by looking at the exponential time constant of the decay in fluorescence. It would be interesting to see if there is a difference in these rates depending on dwell time. For example, in Figure 3A, the fast events (0.1s, 0.3s, 2.4s) appear to acidify faster than the slow events. If it holds for the larger data set, then this would be a new finding.

The reviewer raises an important point. Previous work has shown that increasing extracellular Ca^2+^ concentration not only slows down endocytosis (longer dwell times) but also slows down reacidification (measured using the exponential time constant of the decay in fluorescence) (see Leitz and Kavalali, 2011 and Li et al., 2017). Strikingly, the duration of dwell time and the speed of reacidification did not show a correlation for the data analyzed in the present work (see Author response image 1), although we could detect a similar Ca^2+^-dependent slow down in dwell and decay times as before (data not shown). This result points to potential differences in the nature and size of the endocytic organelle generated after retrieval. We prefer to address this issue in more depth in future studies with additional methods. Although, in the current manuscript, we focused on endocytosis speed, a preliminary analysis revealed fluorescence decay time constants of ~1.0-1.25 s, in agreement with the values proposed by Budzinski and colleagues (ACS Chem Neurosci, 2011), suggesting endocytic organelles of a size in the order of a synaptic vesicle.

**Author response image 1. respfig1:** a and b **–** Cumulative histograms of single vesicle fusion event amplitudes at room temperature (24 °C) and physiological temperature (34 °C), measured with vGluT1-pHluorin. c and d **–** Plot of decay time constant as a function of dwell time duration at room temperature (24 °C) and physiological temperature (34 °C), measured with vGluT1-pHluorin. Solid black lines: linear fit, showing no correlation (R^2^ ~ 0). e **–** Cumulative distribution of decay time constants for single vesicle fusion events showing no differences at the two measured temperatures. Inset: mean value ± SEM. N of experiments are indicated in Figure 3 legend of the main manuscript.

The authors state that overexpression of V0a1-pHluorin results in a "small" increase in mIPSC frequency. Actually, the figure shows a 2-fold increase. Also, does overexpression have an effect on evoked IPSCs?

Reviewers are correct, the increase in mIPSC is 2-fold on average, we have corrected the text indicating that “V0a1-pHluorin expression (…) caused a 2-fold increase in miniature inhibitory postsynaptic currents (mIPSC) frequency, with no changes in amplitude”. Overexpression of either V0a1 or V0a1-pHluorin does not alter evoked IPSC amplitude or charge transfer, and paired-pulse ratio is also unaffected, as shown in Figure 6—figure supplement 1A to D.

The authors suggest that the results in Figure 6L differ from those in Figure 3. Please provide statistics to assess if the differences are significant.

Since the pie charts in figure 6L represent the percentage of each group of endocytosis events with respect to the total, it is not possible to perform statistics to compare this distribution with the ones presented in Figures 3 or 4. Moreover, the total number of events analyzed in figure 6J-L is smaller compared to the other probes. Therefore, as the reviewer pointed out, it is hard to confidently state that “the results in figure 6L differ from those in figure 3”, we decided to remove that sentence from the text.